# A Scalable MIP-based Method for Learning Optimal Multivariate Decision Trees

**Haoran Zhu, Pavankumar Murali, Dzung T. Phan, Lam M. Nguyen, Jayant R. Kalagnanam**
IBM Research, Thomas J. Watson Research Center, Yorktown Heights, NY 10598, USA
`haoran@ibm.com, pavanm@us.ibm.com, phandu@us.ibm.com,`
`LamNguyen.MLTD@ibm.com, jayant@us.ibm.com`

## Abstract

Several recent publications report advances in training optimal decision trees (ODT) using mixed-integer programs (MIP), due to algorithmic advances in integer programming and a growing interest in addressing the inherent suboptimality of heuristic approaches such as CART. In this paper, we propose a novel MIP formulation, based on a 1-norm support vector machine model, to train a multivariate ODT for classification problems. We provide cutting plane techniques that tighten the linear relaxation of the MIP formulation, in order to improve run times to reach optimality. Using 36 data-sets from the University of California Irvine Machine Learning Repository, we demonstrate that our formulation outperforms its counterparts in the literature by an average of about 10% in terms of mean out-of-sample testing accuracy across the data-sets. We provide a scalable framework to train multivariate ODT on large data-sets by introducing a novel linear programming (LP) based data selection method to choose a subset of the data for training. Our method is able to routinely handle large data-sets with more than 7,000 sample points and outperform heuristics methods and other MIP based techniques. We present results on data-sets containing up to 245,000 samples. Existing MIP-based methods do not scale well on training data-sets beyond 5,500 samples.

## 1  Introduction

Decision tree models have been used extensively in machine learning (ML), mainly due to their transparency which allows users to derive interpretation on the results. Standard heuristics, such as CART [6], ID3 [17] and C4.5 [18], help balance gains in accuracy with training times for large-scale problems. However, these greedy top-down approaches determine the split at each node one-at-a-time without considering future splits at subsequent nodes. This means that splits at nodes further down in the tree might affect generalizability due to weak performance. Pruning is typically employed to address this issue, but this means that the training happens in two steps – first, top-down training and then pruning to identify better (stronger) splits. A better approach would be a one-shot training of the entire tree that determines splits at each node with full knowledge of all future splits, while optimizing, say, the misclassification rate [4]. The obtained decision tree is usually referred to as an *optimal decision tree (ODT)* in the literature.

The discrete nature of decisions involved in training a decision tree has inspired researchers in the field of Operations Research to encode the process using a mixed-integer programming (MIP) framework [5, 12, 19, 22, 23]. This has been further motivated by algorithmic advances in integer optimization [4]. An MIP-based ODT training method is able to learn the entire decision tree in a single step, allowing each branching rule to be determined with full knowledge of all the remaining rules. Papers on ODT have used optimality criteria such as the average testing length [14], training accuracy, the combination of training accuracy and model interpretability [4, 13], and the combination of training

accuracy and fairness [1]. In a recent paper, Aghaei et al. [2] propose a strong max-flow based MIP formulation to train univariate ODT for binary classification. The flexibility given by the choice of different objective functions, as well as linear constraints in the MIP model, also allows us to train optimal decision trees under various optimality criteria.

Yet another drawback of heuristic approaches is the difficulty in training multivariate (or oblique) decision trees, wherein splits at a node use multiple variables, or hyperplanes. While multivariate splits are much stronger than univariate (or axis-parallel) splits, they are more complicated because the splitting decision at each node cannot be enumerated. Many approaches have been proposed to train multivariate decision trees including the use of support vector machine (SVM) [3], logistic regression [21] and Householder transformation [25]. As noted in [4], these approaches do not perform well on large-scale data-sets as they also rely on top-down induction for the training process.

The *first contribution* of this paper is a new MIP formulation, SVM1-ODT, for training multivariate decision trees. Our formulation differs from others in the literature – we use a 1-norm SVM to maximize the number of correctly classified instances and to maximize the margin between clusters at the leaf nodes. We show that this formulation produces ODTs with a higher out-of-sample accuracy compared to the ODTs trained from state-of-the-art MIP models and heuristic methods on 20 data-sets selected from the UCI ML repository. In Sect. 4, we report the testing performance of the ODT trained using our formulation and show that is has an average improvement of 6-10% in mean out-of-sample accuracy for a decision tree of depth 2, and an average improvement of 17-20% for a decision tree of depth 3.

The *second contribution* of this paper is towards increasing tractability of the MIP-based ODT training for very large data-sets that are typical in real-world applications. It is imperative to bear in mind that the tractability of MIPs limits the size of the training data that can be used. Prior MIP-based ODT training formulations [4, 23] are intractable for large-sized data-sets (more than 5000 samples) since the number of variables and constraints increase linearly with the size of the training data. We address tractability in two steps. First, we tighten the LP-relaxation of SVM1-ODT by providing new cutting planes and getting rid of the big-M constant. Second, we propose an efficient linear programming (LP) -based data-selection method to be used prior to training the tree. This step is comprised of selecting a subset of data points that maximizes the information captured from the entire data-set.

Our method is able to routinely handle large data-sets with more than 7,000 sample points. We present results on data-sets containing up to 245,000 samples. Existing MIP-based methods do not scale well on training data beyond 5,000 samples, and do not provide a significant improvement over a heuristic approach. For large-scale data-sets, when SVM1-ODT is used along with the LP-based data selection method, our results indicate that the resulting decision trees offer higher training and testing accuracy, compared to CART (see Sect. 4, Figure 2). However, solely using any MIP-based formulation (including ours) without data selection can rarely outperform CART, due to the model becoming intractable resulting in the MIP solver failing to find any better feasible solution than the initially provided warm-start solution. This indicates that any loss of information from data-selection is more than adequately compensated by the use of optimal decision trees (using the SVM1-ODT formulation).

## 2   MIP formulation for training multivariate ODT for classification

In this section, we present our formulation to train an optimal multivariate classification tree using a data-set comprising numerical features, and for general data-sets containing categorical features, we propose an extension of such formulation in the supplementary material. For any $n \in \mathbb{Z}_+$, let $[n] := \{1, 2, \ldots, n\}$ denote a finite set of data points, $[Y] = \{1, 2, \ldots, Y\}$ be a set of class labels, and $[d] = \{1, 2, \ldots, d\}$ be the index set of all features. Our formulation is established for the balanced binary tree with depth $D$. Let the set of *branch nodes* of the tree be denoted by $\mathcal{B} := \{1, \ldots, 2^D - 1\}$, and the set of *leaf nodes* be denoted by $\mathcal{L} := \{2^D, \ldots, 2^{D+1} - 1\}$. Similar to [4], let $A_R(l)$ and $A_L(l)$ denote the sets of ancestors of leaf node $l$ whose right and left branches, respectively, are on the path from the root node to leaf node $l$.

Next, we define the variables to be used. Each data point $i \in [n]$ is denoted by $(\mathbf{x}_i, y_i)$, where $\mathbf{x}_i$ is a $d$-dimensional vector, and $y_i \in [Y]$. Since we train multivariate trees, we use a branching hyperplane at branch node $b$, denoted by $\langle \mathbf{h}_b, \mathbf{x}_i \rangle = g_b$, where $g_b$ is the bias term in the hyperplane. Indicator binary variable $c_i = 1$ when $i$ is misclassified and 0 otherwise. Indicator binary variable $e_{il} = 1$

when $i$ enters leaf node $l$. Variable $\hat{y}_i \in [Y]$ denotes the predicted label for $i$, and the decision variable $u_l \in [Y]$ is the label assigned to leaf node $l$. We let $m_{ib}$ denote the slack variable for the soft margin for each point $i$ corresponding to a hyperplane $\langle \mathbf{h}_b, \mathbf{x}_i \rangle = g_b$ used in the SVM-type model (3). The objective for the learning problem shown in (1) attempts to minimize the total misclassification ($\sum_i c_i$), the 1-norm SVM margin ($\sum_b \|\mathbf{h}_b\|_1$) and the sum of slack variables for classification ambiguity subtracted from the soft margin ($\sum_{i,b} m_{ib}$). Additionally, $\sum_b \|\mathbf{h}_b\|_1$ helps promote sparsity in the decision hyperplanes constructed at the branch nodes of a decision tree during the training process.

Then, **SVM1-ODT** can be expressed as follows:

$$\min \quad \sum_{i \in [n]} c_i + \alpha_1 \sum_{i \in [n], b \in \mathcal{B}} m_{ib} + \alpha_2 \sum_{b \in \mathcal{B}} \|\mathbf{h}_b\|_1 \tag{1a}$$

$$\text{s.t.} \quad (y_i - Y)c_i \leq y_i - \hat{y}_i \leq (y_i - 1)c_i, \forall i \in [n] \tag{1b}$$

$$\hat{y}_i = \sum_{l \in \mathcal{L}} w_{il}, \forall i \in [n] \tag{1c}$$

$$w_{il} \geq e_{il}, u_l - w_{il} + e_{il} \geq 1, \forall i \in [n], l \in \mathcal{L} \tag{1d}$$

$$Y e_{il} + u_l - w_{il} \leq Y, w_{il} \leq Y e_{il}, \forall i \in [n], l \in \mathcal{L} \tag{1e}$$

$$g_b - \sum_{j \in [d]} h_{bj} x_{ij} = p_{ib}^+ - p_{ib}^-, \forall i \in [n], b \in \mathcal{B} \tag{1f}$$

$$p_{ib}^+ \leq M(1 - e_{il}), \forall i \in [n], l \in \mathcal{L}, b \in A_R(l) \tag{1g}$$

$$p_{ib}^- + m_{ib} \geq \epsilon e_{il}, \forall i \in [n], l \in \mathcal{L}, b \in A_R(l) \tag{1h}$$

$$p_{ib}^- \leq M(1 - e_{il}), \forall i \in [n], l \in \mathcal{L}, b \in A_L(l) \tag{1i}$$

$$p_{ib}^+ + m_{ib} \geq \epsilon e_{il}, \forall i \in [n], l \in \mathcal{L}, b \in A_L(l) \tag{1j}$$

$$\sum_{l \in \mathcal{L}} e_{il} = 1, \forall i \in [n] \tag{1k}$$

$$\hat{y}_i, w_{il}, h_{bj}, g_b \in \mathbb{R}, p_{ib}^+, p_{ib}^-, m_{ib} \in \mathbb{R}_+, e_{il}, c_i \in \{0, 1\}, u_l \in \{1, \ldots, Y\}. \tag{1l}$$

We notice that $\mathbf{h}_b$, $g_b$ and $u_l$ are main decision variables to characterize a decision tree, while $\hat{y}_i$, $w_{il}$, $p_{ib}^+, p_{ib}^-, m_{ib}, e_{il}$ and $c_i$ are derived variables for the MIP model.

Constraint (1b) expresses the relationship between $c_i$ and $\hat{y}_i$. If $\hat{y}_i = y_i$, then $c_i = 0$ since $c_i$ is minimized. If $\hat{y}_i \neq y_i$ and $\hat{y}_i < y_i$, then $(y_i - 1)c_i \geq y_i - \hat{y}_i \geq 1$, thus $c_i = 1$, Similarly, if $\hat{y}_i > y_i$, then $(y_i - Y)c_i \leq -1$, thus $c_i = 1$. In a balanced binary tree with a fixed depth, the branching rule is given by: $i$ goes to the left branch of $b$ if $\langle \mathbf{h}_b, \mathbf{x}_i \rangle \leq g_b$, and goes to the right side otherwise. Predicted class $\hat{y}_i = \sum_{l \in \mathcal{L}} u_l \cdot e_{il}$. Since $u_l \cdot e_{il}$ is bilinear, we perform McCormick relaxation [15] of this term using an additional variable $w_{il}$ such that $w_{il} = u_l \cdot e_{il}$. Since $e_{il}$ is binary, this McCormick relaxation is exact. That is to say, $\hat{y}_i = \sum_{l \in \mathcal{L}} u_l \cdot e_{il}$ if and only if (1c)-(1e) hold for some extra variables $w$. Since $u_l$ and $e_{il}$ are integer variables, it follows that $\hat{y}_i$ also integral. Constraints (1f), (1g) and (1i) formulate the branching rule at each node $b \in \mathcal{B}$: if $i$ goes to the left branch at node $b$, then $g_b - \sum_{j \in [d]} h_{bj} x_{ij} \geq 0$, and if it goes to the right side, then $g_b - \sum_{j \in [d]} h_{bj} x_{ij} \leq 0$. As per MIP convention, we formulate this relationship by separating $g_b - \sum_{j \in [d]} h_{bj} x_{ij}$ into a pair of complementary variables $p_{ib}^+$ and $p_{ib}^-$ (meaning $p_{ib}^+$ and $p_{ib}^-$ cannot both be strictly positive at the same time), and forcing one of these two variables to be 0 through the big-M method [26]. We should remark that this is not exactly the same as our branching rule: when $i$ goes to the right branch, it should satisfy $g_b - \sum_{j \in [d]} h_{bj} x_{ij} < 0$ strictly. The only special case is when $p_{ib}^+ = p_{ib}^- = 0$. However, due to the two constraints (1h), (1j), and the penalizing term $m_{ib} \geq 0$, this phenomenon cannot occur. Constraint (1k) enforces that each $i$ should be assigned to exactly one leaf node. For a given dataset, model parameters $\epsilon$, $\alpha_1$ and $\alpha_2$ are tuned via cross-validation. In the following section, we provide explanations for constraints (1h), (1j) and objective function (1a), and some strategies to tighten SVM1-ODT.

## 2.1 Multi-hyperplane SVM model for ODT

When constructing a branching hyperplane, we normally want to maximize the shortest distance from this hyperplane to its closest data points. For any branching node associated with a hyperplane $\langle \mathbf{h}_b, \mathbf{x}_i \rangle = g_b$ by fixing other parts of the tree, we can view the process of learning the hyperplane as constructing a binary classifier over data points $\{(\mathbf{x}_i, \overline{y}_i)\}$ that reach the node. The artificial label $\overline{y}_i \in \{\text{left}, \text{right}\}$ $(\equiv \{-1, +1\})$ is derived from the child of the node: $\mathbf{x}_i$ goes to the left or the right; that is determined by $e_{il}$. This problem is reduced to an SVM problem for each branching node. Applying the 1-norm SVM model with soft margin [16, 27] to the node $b$ gives us

$$
\begin{aligned}
\min \quad & \tfrac{1}{2}\|\mathbf{h}_b\|_1 + \alpha_1 \sum_{i \in [n]} m_{ib} \\
\text{s.t.} \quad & \left|g_b - \sum_{j \in [d]} h_{bj} x_{ij}\right| + m_{ib} \geq \epsilon e_{il}, \forall i \in [n], l \in \mathcal{L}_b.
\end{aligned}
\tag{2}
$$

Here $\mathcal{L}_b$ is the set of leaf nodes that have node $b$ as an ancestor and $m_{ib}$ denotes the slack variable for soft margin. We slightly modify the original 1-norm SVM model by using a small constant $\epsilon e_{il}$ in (2) instead of $e_{il}$ to prevent variables from getting too big. The constraint is only active when $e_{il} = 1$, namely data $i$ enters the branching node $b$, and $e_{il} = 1$ implicitly encodes $\overline{y}_i$.

We use 1-norm, instead of the Euclidean norm, primarily because it can be linearized. The sparsity for $\mathbf{h}_b$ targeted heuristically by including the 1-norm term [27] in the objective allows for feature selection at each branching node. As we noted previously, we can express the term $g_b - \sum_{j \in [d]} h_{bj} \mathbf{x}_{ij}$ as the difference of two positive complementary variables, and the absolute value $|g_b - \sum_{j \in [d]} h_{bj} \mathbf{x}_{ij}|$ just equals one of these two complementary variables, depending on which branch such a data point enters. When taking the sum over all branching nodes, we have the following *multi-hyperplane SVM* problem to force data points close to the center of the corresponding sub-region at each leaf node $l \in \mathcal{L}$:

$$
\begin{aligned}
\min \quad & \sum_{b \in A_L(l) \cup A_R(l)} \tfrac{1}{2}\|\mathbf{h}_b\|_1 + \alpha_1 \sum_{i \in [n], b \in A_L(l) \cup A_R(l)} m_{ib} \\
\text{s.t.} \quad & \left|g_b - \sum_{j \in [d]} h_{bj} x_{ij}\right| + m_{ib} \geq \epsilon e_{il}, \\
& \forall i \in [n], b \in A_L(l) \cup A_R(l).
\end{aligned}
\tag{3}
$$

Note that $\bigcup_{l \in \mathcal{L}} (A_L(l) \cup A_R(l)) = \mathcal{B}$. By combining (3) over all leaf nodes $l \in \mathcal{L}$ we obtain:

$$
\begin{aligned}
\min \quad & \sum_{b \in \mathcal{B}} \tfrac{1}{2}\|\mathbf{h}_b\|_1 + \alpha_1 \sum_{i \in [n], b \in \mathcal{B}} m_{ib} \\
\text{s.t.} \quad & (1f) - (1j).
\end{aligned}
\tag{4}
$$

Adding the misclassification term $\sum_i c_i$ back into the objective function, and assigning some regularization parameters, we end up getting the desired objective function in (1a).

In LP, the minimized absolute value term $|h_{bj}|$ can be easily formulated as an extra variable $h'_{bj}$ and two linear constraints: $h'_{bj} = h^+_{bj} + h^-_{bj}, h_{bj} = h^+_{bj} - h^-_{bj}, h^+_{bj}, h^-_{bj} \geq 0$. We highlight the major difference here from another recent work [4] in using MIP to train ODT. First, in their formulation (OCT-H), they consider penalizing the number of variables used across all branch nodes in the tree, in order to encourage model simplicity. Namely, their minimized objective is: $\sum_i c_i + \alpha_1 \sum_b \|\mathbf{h}_b\|_0$. However, this requires some additional binary variables for each $h_{bj}$, which makes the MIP model even harder to solve. In fact, there is empirical evidence that using the 1-norm helps with model sparsity (e.g., LASSO [20], 1-norm SVM [27]). For that reason, we do not bother adding another 0-norm regularization term into our objective. Secondly, unlike in [4], we use variables $u_l$ to denote the assigned label on each leaf, where $u_l \in [Y]$. The next theorem shows, in SVM1-ODT (1), integer $\mathbf{u}$ variables can be equivalently relaxed to be continuous between $[1, Y]$. This relaxation is important in terms of optimization tractability.

**Theorem 1.** *Every integer variable $u_l, l \in \mathcal{L}$, in (1) can be relaxed to be continuous in $[1, Y]$.*

We note that all proofs are delegated to the supplementary material.

## 2.2 Strategies to tighten MIP formulation

For MIPs, it is well-known that avoiding the use of a big-M constant can enable us to obtain a tighter linear relaxation of the formulation, which helps an optimization algorithm such as branch-and-bound converge faster to a global solution. The next theorem shows that the big-M constant $M$ can be

fixed to be 1 in (1) for SVM1-ODT, with the idea being to disperse the numerical issues between parameters by re-scaling. The numerical instability for a very small $\epsilon$, as in the theorem below, should be easier to handle by an MIP solver than a very big $M$.

**Theorem 2.** *Let* $(\mathbf{h}'_b, g'_b)_{b \in \mathcal{B}}, (u'_l)_{l \in \mathcal{L}}, (\hat{y}'_i, c'_i)_{i \in [n]}, (w'_{il}, e'_{il})_{i \in [n], l \in \mathcal{L}}, (p_{ib}^{+\prime}, p_{ib}^{-\prime}, m'_{ib})_{i \in [n], b \in \mathcal{B}}$ *be a feasible solution to* (1) *with parameters* $(\alpha_1, \alpha_2, M, \epsilon)$. *Then, it is also an optimal solution to* (1) *with parameters* $(\alpha_1, \alpha_2, M, \epsilon)$, *if and only if* $(\mathbf{h}'_b/M, g'_b/M)_{b \in \mathcal{B}}, (u'_l)_{l \in \mathcal{L}}, (\hat{y}'_i, c'_i)_{i \in [n]}, (w'_{il}, e'_{il})_{i \in [n], l \in \mathcal{L}}, (p_{ib}^{+\prime}/M, p_{ib}^{-\prime}/M, m'_{ib}/M)_{i \in [n], b \in \mathcal{B}}$ *is an optimal solution to* (1) *with parameters* $(M\alpha_1, M\alpha_2, 1, \epsilon/M)$.

Note that $\{(\mathbf{h}'_b, g'_b)_{b \in \mathcal{B}}, (u'_l)_{l \in \mathcal{L}}\}$ and $\{(\mathbf{h}'_b/M, g'_b/M)_{b \in \mathcal{B}}, (u'_l)_{l \in \mathcal{L}}\}$ represent the same decision tree, since $\langle \mathbf{h}'_b, \mathbf{h} \rangle \le g'_b$ is equivalent to $\langle \mathbf{h}'_b/M, \mathbf{x} \rangle \le g'_b/M$, at each branch node $b \in \mathcal{B}$.

In MIP parlance, a *cutting-plane* (also called a *cut*) is a linear constraint that is not part of the original model and does not eliminate any feasible integer solutions. Pioneered by Gomory [10, 11], cutting-plane methods, as well as branch-and-cut methods, are among the most successful techniques for solving MIP problems in practice. Numerous types of cutting-planes have been studied in integer programming literature and several of them are incorporated in commercial solvers (see, e.g., [7, 26]). Even though the state-of-the-art MIP solvers can automatically generate cutting-planes during the solving process, these cuts usually do not take the specific structure of the model into consideration. Therefore, most commercial solvers allow the inclusion of user cuts, which are added externally by users in order to further tighten the MIP model. In this section, we propose a series of cuts for SVM1-ODT, and they are added once at the beginning before invoking a MIP solver. For the ease of notation, we denote by $\mathcal{N}_k$ the set of data points with the same dependent variable value $k$, i.e., $\mathcal{N}_k := \{i \in [n] : y_i = k\}$, for any $k \in [Y]$.

**Theorem 3.** *Given a set* $I \subseteq [n]$ *with* $|I \cap \mathcal{N}_k| \le 1$ *for any* $k \in [Y]$. *Then for any* $L \subseteq \mathcal{L}$, *the inequality*

$$\sum_{i \in I} c_i \ge \sum_{i \in I, l \in L} e_{il} - |L| \tag{5}$$

*is a valid cutting-plane for SVM1-ODT* (1).

Here the index set $I$ is composed by arbitrarily picking at most one data point from each class $\mathcal{N}_k$.

Theorem 3 admits $\Omega(n^Y \cdot 2^{2^D})$ number of cuts. Next, we list cuts that will be added later as user cuts for our numerical experiments. They all help provide lower bounds for the term $\sum_{i \in [n]} c_i$, which also appear in the objective function of our SVM1-ODT formulation. Our cutting planes are added once at the beginning before invoking a MIP solver.

**Proposition 1.** *Let* $\{|\mathcal{N}_k| \mid k \in [Y]\} = \{s_1, \ldots, s_k\}$ *with* $s_1 \le s_2 \le \ldots \le s_Y$. *Then the following inequalities*

1. $\forall l \in \mathcal{L}, \sum_{i \in [n]} c_i \ge \sum_{i \in [n]} e_{il} - s_Y$;

2. $\forall l \in \mathcal{L}, \sum_{i \in [n]} (c_i + e_{il}) \ge s_1 \cdot (Y - 2^D + 1)$;

3. $\sum_{i \in [n]} c_i \ge s_1 + \ldots + s_{Y-2^D}$ *if* $Y > 2^D$.

*are all valid cutting-planes for SVM1-ODT* (1).

Note that the second inequality is only added when $Y \ge 2^D$, and the last lower bound inequality is only added when $Y > 2^D$. As trivial as the last lower bound inequality might seem, in some cases it can be quite helpful. During the MIP solving process, when the current best objective value meets this lower bound value, optimality can be guaranteed and the solver will terminate the branch-and-bound process. Therefore, a tightness of the lower bound has a significant impact on run time.

## 3  LP-based data-selection procedure

As mentioned previously, our main focus is to be able to train ODT over very large training data-sets. For the purpose of scalability, we rely on a data-selection method prior to the actual training process using SVM1-ODT.

The outline for our procedure is as follows: first, we use a decision tree trained using a heuristic (e.g., CART) as an initial solution. Next, data-selection is performed on clusters represented by the data points with the same dependent values at each leaf node. Finally, we merge all the data subsets selected from each cluster as the new training data, and use SVM1-ODT to train a classification tree on this data-set. In each cluster, our data-selection is motivated by the following simple heuristic: suppose for a data subset $I_0$ all the points in $\text{conv}\{\mathbf{x}_i \mid i \in I_0\}$ are correctly classified as label $y$. Then, we can drop out all the data points that lie in the interior of $\text{conv}\{\mathbf{x}_i \mid i \in I_0\}$ from our training set, since by assigning $\{\mathbf{x}_i \mid i \in I_0\}$ to the same leaf node and labeling it with $y$, we will also correctly classify all the remaining data points inside their convex combination. With that in mind, a data subset $I_0$ is selected as per the following two criteria: (1) the points within the convex hull $\text{conv}\left(\{\mathbf{x}_i \mid i \in I_0\}\right)$ are as many as possible; and (2) $|I_0|$ is as small as possible. In each cluster $\mathcal{N} \subseteq [n]$, the following 0-1 LP can be defined to do data-selection:

$$
\begin{aligned}
\min \quad & \mathbf{f}^T \mathbf{a} - \mathbf{g}^T \mathbf{b} \\
\text{s.t.} \quad & -\epsilon' \cdot \mathbf{1} \leq b_j \mathbf{x}_j - \sum_{i \neq j} \lambda_{ji} \mathbf{x}_i \leq \epsilon' \cdot \mathbf{1}, \forall j \in \mathcal{N} \\
& \sum_{i \neq j} \lambda_{ji} = b_j, \forall j \in \mathcal{N} \\
& 0 \leq \lambda_{ji} \leq a_i, \forall i \neq j \in \mathcal{N} \\
& a_j + b_j \leq 1, \forall j \in \mathcal{N} \\
& a_j, b_j \in \{0, 1\}, \forall j \in \mathcal{N}.
\end{aligned}
\tag{6}
$$

Here $\mathbf{f}, \mathbf{g}$ are two parameter vectors with non-negative components. Data point $\mathbf{x}_i$ is selected if $a_i = 1$. Data point $\mathbf{x}_j$ is contained in the convex combination of selected data points if $b_j = 1$. When $\epsilon' = 0$, for any $j \in \mathcal{N}$ with $b_j = 1$, the first two constraints express $\mathbf{x}_j$ as the convex combination of points in $\{\mathbf{x}_i \mid \lambda_{ji} > 0\}$. Here we introduce a small constant $\epsilon'$ to allow some perturbation. The third inequality $0 \leq \lambda_{ji} \leq a_i$ means we can only use selected data points, which are those with $a_i = 1$, to express other data points. The last constraint $a_j + b_j \leq 1$ ensures that any selected data point cannot be expressed as a convex combination of other selected data points. Depending on the choice of $\mathbf{f}, \mathbf{g}$ and $\epsilon'$, we have many different variants of (6). In the next section, we describe one variant of data-selection. We discuss *balanced* data-selection in the supplementary material.

## 3.1 Selecting approximal extreme points

We notice that the original 0-1 LP can be formulated to maximize the number of data points inside the convex hull of selected data points by selecting $\mathbf{f} = \mathbf{0}$ and $\mathbf{g} = \mathbf{1}$. This special case of (6) is used because choosing these values allows us to decompose it into $N$ smaller LPs while maximizing the points inside the convex hull. By projecting out variable $\mathbf{a}$, the resulting 0-1 LP is equivalent to the following LP, as shown by the next result:

$$
\begin{aligned}
\max \quad & \sum_{i \in \mathcal{N}} b_i \\
\text{s.t.} \quad & -\epsilon' \cdot \mathbf{1} \leq b_j \mathbf{x}_j - \sum_{i \neq j} \lambda_{ji} \mathbf{x}_i \leq \epsilon' \cdot \mathbf{1}, \forall j \in \mathcal{N} \\
& \sum_{i \neq j} \lambda_{ji} = b_j, \forall j \in \mathcal{N} \\
& 0 \leq b_j \leq 1, \forall j \in \mathcal{N}.
\end{aligned}
\tag{7}
$$

We note that such an LP is decomposable: it can be decomposed into $|\mathcal{N}|$ many small LPs, each with $d + 2$ constraints and $|\mathcal{N}|$ variables, and each can be solved in parallel.

**Theorem 4.** *The following hold.*

1) *If $\epsilon' = 0$, then for any optimal solution $(\boldsymbol{b}, \bar{\lambda})$ of (7), there exists $\lambda$ s.t. $(\boldsymbol{b}, \lambda)$ is optimal solution of (6) with $\boldsymbol{f} = \boldsymbol{0}, \boldsymbol{g} = \boldsymbol{1}$, and vice versa;*

2) *If $\epsilon' > 0$, then for any optimal solution $(\boldsymbol{b}, \bar{\lambda})$ of (7), there exists $\lambda$ s.t. $(\lfloor \boldsymbol{b} \rfloor, \lambda)$ is an optimal solution of (6) with $\boldsymbol{f} = \boldsymbol{0}, \boldsymbol{g} = \boldsymbol{1}$. Here, $\lfloor \boldsymbol{b} \rfloor$ is a vector with every component being $\lfloor b_i \rfloor$.*

## 3.2 Data-selection algorithm

For each cluster $\mathcal{N}$, let $I_{\mathcal{N}} := \{j \in \mathcal{N} : b_j = 1\}$. Note that among those extreme points in $\{i \in \mathcal{N} : \exists j \in I_{\mathcal{N}}, \text{ s.t. } \lambda_{ji} > 0\}$, some of them may be outliers, which will result in $\lambda_{ji}$ being a small number for all $j \in \mathcal{N}$. From the classic Carathéodory's theorem in convex analysis, for any point $\mathbf{x}_j$ with $j \in I_{\mathcal{N}}$, it can be written as the convex combination of at most $d + 1$ extreme points. Then in that case, there must exist $i$ with $\lambda_{ji} \geq \frac{1}{d+1}$. Denote $J_{\mathcal{N}} := \{i \in \mathcal{N} : \exists j \in I_{\mathcal{N}}, \text{ s.t. } \lambda_{ji} \geq$

$\frac{1}{d+1}\}$, $K_{\mathcal{N}} := \mathcal{N} \setminus (I_{\mathcal{N}} \cup J_{\mathcal{N}})$. One can show that the size of set $J_{\mathcal{N}}$ is upper bounded by $(d+1)|I_{\mathcal{N}}|$, so if $|I_{\mathcal{N}}|$ is small, $|J_{\mathcal{N}}|$ is also relatively small. For those data points in $K_{\mathcal{N}}$, we use a simple heuristic for data-selection inspired by support vector machines, wherein the most critical points for training are those closest to the classification hyperplane. That is, for some data-selection threshold number $N$ and the boundary hyperplanes $h \in H$ (hyperplanes for each leaf node), we sort data point $i \in K_{\mathcal{N}}$ by $\min_{h \in H} \text{dist}(\mathbf{x}_i, h)$, in an increasing order, and select the first $N$ points. This heuristic is called **hyperplane based data-selection** and is presented in Algorithm 1 below. The LP in (7) above is solved in the second step of Algorithm 1. When $|I_{\mathcal{N}}|$ is relatively large, we can simply select all the extreme points as the new training data. When $|I_{\mathcal{N}}|$ is relatively small, we can select those points with index in $J_{\mathcal{N}}$ as the new training data. The remaining data points in $K_{\mathcal{N}}$ are selected according to the above hyperplane based data-selection heuristic.

---

**Algorithm 1** LP-based data-selection in each cluster $\mathcal{N}$

---

**Given** $\beta_1, \beta_2 \in (0, 1), \beta_2 < (d+1)(1 - \beta_1)$;
**Solve** LP (7) and obtain the optimal solution $(\mathbf{b}, \bar{\lambda})$,
denote $I_{\mathcal{N}} := \{i \in \mathcal{N} : b_i = 1\}, \lambda = T(\bar{\lambda})$,
$J_{\mathcal{N}} := \{i \in \mathcal{N} : \exists j \in I_{\mathcal{N}}, \text{ s.t. } \lambda_{ji} \geq \frac{1}{d+1}\}$,
$K_{\mathcal{N}} := \mathcal{N} \setminus (I_{\mathcal{N}} \cup J_{\mathcal{N}})$;
**if** $|I_{\mathcal{N}}|/|\mathcal{N}| \geq 1 - \beta_1$ **then**
    Select $\mathcal{N} \setminus I_{\mathcal{N}}$ as training set;
**else if** $|J_{\mathcal{N}}| > \beta_2|\mathcal{N}|$ **then**
    Select $J_{\mathcal{N}}$ as training set;
**else**
    For $K_{\mathcal{N}}$, do **Hyperplane Data-selection** and pick the first $\beta_2|\mathcal{N}| - |J_{\mathcal{N}}|$ points, together with $J_{\mathcal{N}}$, as the selected new training set.
**end if**

---

In Algorithm 1, $T(\bar{\lambda})$ is the transformation we used in the proof of Theorem 4 to construct a feasible solution for (6), and $\beta_1, \beta_2$ are some pre-given threshold parameters. For MIP solvers such as CPLEX, the obtained optimal solution $(\mathbf{b}, \bar{\lambda})$ is a vertex in the associated polyhedron. In that case, $T(\bar{\lambda}) = \bar{\lambda}$.

For large data-sets, for (7), we observed that it took a considerable amount of time to import the constraints into the LP solver and solve it. However, since LP (7) can be decomposed into $|\mathcal{N}|$ number of much smaller LPs, the computational process can be accelerated dramatically.

## 4   Numerical experiments

We present results mainly from two types of numerical experiments to evaluate the performance of our ODT training procedure: (1) benchmark the mean out-of-sample performance of the ODT trained using SVM1-ODT on medium-sized data-sets ($n \leq 7000$), w.r.t. its counterparts in literature; and (2) benchmark the mean out-of-sample performance of the ODT trained using SVM1-ODT together with our data-selection procedure on large-scale data-sets ($7,000 \leq n \leq 245,000$), w.r.t. CART and OCT-H [4]. For benchmarking, we use data-sets from the UCI Machine Learning Repository [9].

**Accuracy of multivariate ODT:** We tested the accuracy of the ODT trained using our SVM1-ODT against baseline methods CART and OCT-H [4].

We used the same training-testing split as in [4], which is 75% of the entire data-set as training set, and the rest 25% as testing set, with 5-fold cross-validation. The time limit was set to be 15 or 30 minutes for medium-sized data-sets, and for larger data-sets we increased it up to 4 hours to ensure that the solver could make sufficient progress. Due to intractability of MIP models and the loss in interpretability for deep decision trees we only train ODT for small tree depths, similar to [4, 12, 23]. With the exception of Table 1, all results shown in this section and in the supplementary material are for $D = 2$.

For the following numerical results, our SVM1-ODT formulation is abbreviated as "S1O", "OCT-H" is the MIP formulation to train multivariate ODT in [4], "Fair" is the MIP formulation from [1] without the fairness term in objective, and "BinOCT" is from [23]. We implemented all these MIP approaches in Python 3.6 and solved them using CPLEX 12.9.0 [8]. We invoked the *DecisionTreeClassifier*

Table 1: Performance on data-sets with more than 4 classes using $D = 3$. The numbers inside the bracket '()' for CART and OCT-H are the numerical results reported in [4].

| | tree depth $D = 3$ | | |
|---|---|---|---|
| data-set | Dermatology | Heart-disease | Image |
| $n$ | 358 | 297 | 210 |
| $d$ | 34 | 13 | 19 |
| $Y$ | 6 | 5 | 7 |
| testing accuracy (%) | | | |
| S1O | **98.9** | **65.3** | **85.7** |
| CART | 76.1(78.7) | 55.6 (54.9) | 57.1 (52.5) |
| OCT-H | 82.6 (83.4) | 56.2 (54.7) | 59.4 (57.4) |
| Fair | 86.4 | 47.2 | 63.3 |
| training accuracy (%) | | | |
| S1O | **100** | 90.2 | **100** |
| CART | 80.7 | 68.0 | 57.1 |
| OCT-H | 89.4 | 81.9 | 82.7 |
| Fair | **100** | **92.3** | **100** |

implementation from scikit-learn to train a decision tree using CART, using default parameter settings. For all methods, the maximum tree depth was set to be the same as our SVM1-ODT.

In Figure (1a), we compare the mean out-of-sample accuracy of the ODT trained from several different MIP formulations. Here the labels on the x-axis represent the names of the data-sets, followed by the data size. In Figure (1b), we have more comparison over the counterparts in literature, along with the BinOCT from [23]. Table 2 shows detailed results about the ODT training, including the running time and training accuracy of different methods. Moreover, we also tested a few data-sets with more than 4 classes using tree depth 3, with time limit being 30 minutes, the results are more exciting: For our tested 3 data-sets, we obtained 20.4 percentage points average improvement over CART, 17.2 percentage points improvement over OCT-H, and 17.7 percentage points improvement over Fair. We list the detailed results in Table 1.

**Training multivariate ODT on large data-sets:** We use SVM1-ODT together with the LP-based data-selection (denoted by "S1O-DS") to train multivariate ODT on large data-sets, which has never been attempted by prior MIP-based training methods in the literature. The time limit for these experiments is 4 hours. Figure 2 depicts the performance of our method w.r.t. CART and OCT-H. For solving S1O and OCT-H, we use the decision tree trained using CART as a warm start solution for CPLEX, as in [4, 24]. For OCT-H, we observe that within the time limit, the solver is either not able to find any new feasible solution other than the one provided as warm start, implying the decision tree trained using OCT-H has the same performance as the one from CART, or the solver simply fails to construct the MIP formulation, which is depicted by the missing bars in the Figure 2. This figure basically means that solely relying on any MIP formulation to train ODT using large data-sets will result in no feasible solution being found within the time limit.

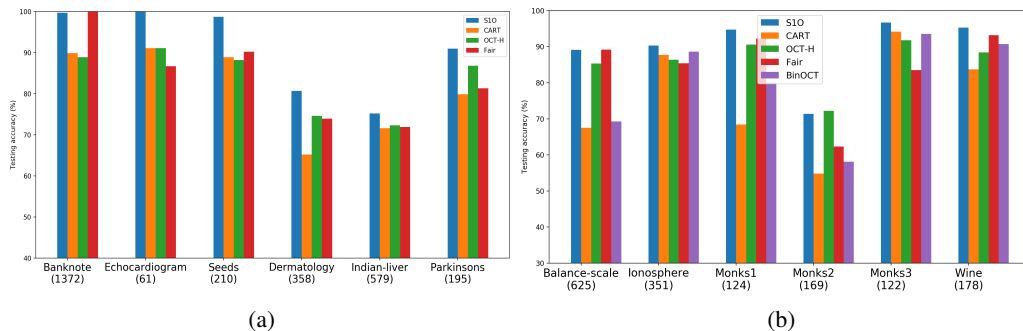

(a)  (b)

Figure 1: Accuracy comparison for multivariate ODT trained on medium-sized data-sets w.r.t (1a): OCT-H and Fair ; (1b): OCT-H, Fair and BinOCT [23], $D = 2$. In-depth results in Tables 3 and 5 in supplementary material.

Table 2: Accuracy and running time on medium-sized data-sets, for $D$=2. The numbers inside the bracket '()' for CART and OCT-H are the numerical results reported in [4].

| data-set | Iris | Congress | Spectf-heart | Breast-cancer | Heart-disease | Image | Hayes-roth |
|---|---|---|---|---|---|---|---|
| $n$ | 150 | 232 | 80 | 683 | 297 | 210 | 132 |
| $d$ | 4 | 16 | 44 | 9 | 13 | 19 | 4 |
| $Y$ | 3 | 2 | 2 | 2 | 5 | 7 | 3 |
| testing accuracy (%) | | | | | | | |
| S1O | **98.6** | **98.0** | **83.3** | **97.6** | **69.9** | **55.0** | 71.9 |
| CART | 92.4 | 93.5 | 72.0 | 91.1 | 57.5 | 42.9 | 55.8 |
| | (92.4) | (98.6) | (69.0) | (92.3) | (54.1) | (38.9) | (52.7) |
| OCT-H | 94.4 | 94.8 | 75.0 | 96.1 | 56.7 | 49.8 | 61.2 |
| | (95.1) | (98.6) | (67.0) | (97.0) | (54.7) | (49.1) | (61.2) |
| Fair | 90.0 | 91.4 | 57.0 | 95.4 | 56.7 | 46.9 | **72.2** |
| training accuracy (%) | | | | | | | |
| S1O | 98.9 | 98.5 | 85 | 97.3 | **75.3** | **56.7** | 76.8 |
| CART | 96.4 | 96.3 | 88.0 | 93.4 | 58.8 | 42.9 | 60.0 |
| OCT-H | 99.5 | 96.2 | 92.5 | 95.3 | 60.5 | 48.0 | 77.4 |
| Fair | **100** | **100** | **100** | **99.7** | 68.4 | 56.6 | **82.8** |
| running time ($s$) | | | | | | | |
| S1O | 8.17 | 1.48 | 0.42 | 517.4 | 900 | 900 | 900 |
| OCT-H | 1.56 | 727.3 | 900 | 900 | 900 | 900 | 900 |
| Fair | 11.14 | 1.82 | 0.23 | 803.5 | 900 | 713.7 | 900 |

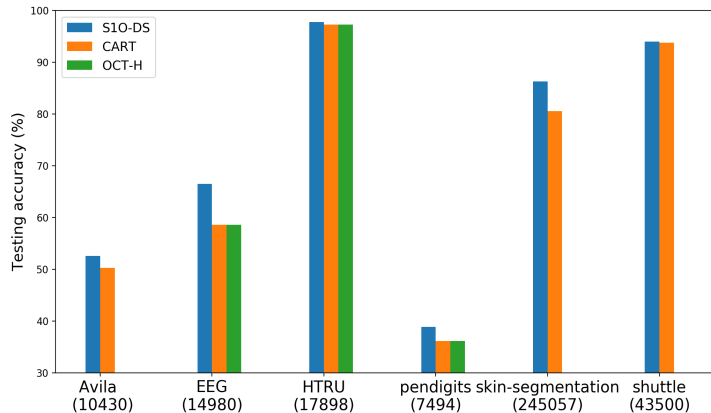

Figure 2: Comparison for large data-sets, $D$=2. In-depth results in Table 7 in supplementary material.

# 5 Conclusions

We propose a novel MIP-based method to train an optimal multivariate classification tree, which has better generalization behavior compared to state-of-the-art MIP-based methods. Additionally, in order to train ODT on very large data-sets, we devise an LP-based data-selection method. Numerical experiments suggest that the combination of these two can enable us to obtain a decision tree with better out-of-sample accuracy than CART and other comparable MIP-based methods, while solely using any MIP-based training method will fail to do that almost certainly. In the current setup, data selection occurs prior to training using SVM1-ODT. So, once a data subset has been selected, it is used at every branch node to determine optimal branching rules. A natural extension of this methodology could be a combined model for ODT training and data selection, wherein the branching rules learned at the each layer, is applied to the entire data-set and data selection is redone at every node in the subsequent layer prior to branching.

## Broader impact

Ensemble methods such as random forests and gradient boosting methods such as XGBoost, and LightGBM typically perform well, in terms of scalability and out-of-sample accuracy, for large-scale classification problems. However, these methods suffer from low interpretability and are incapable of modeling fairness. We hope the scalable MIP-based framework proposed in this paper proves to be seminal in addressing applicability of ODTs to large-scale real-world problems, while relying on the decision tree structure to preserve interpretability. The MIP framework might especially come in handy for sequential or joint prediction-optimization models, wherein the problem structure could be utilized to devise decomposition-based solution procedures. Alternatively, the proposed approach could be used to train a classification tree to provide insights on the behavior of a black-box model.

## Funding statement

The first author of this paper was funded by IBM Research during his internship there.

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
