[Supplementary Material]

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

{1} \le b_j \mathbf{x}_j - \sum_{i \ne j} \lambda_{ji} \mathbf{x}_i \le \epsilon' \cdot \mathbf{1}, \forall j \in \mathcal{N} \\
& \sum_{i \ne j} \lambda_{ji} = b_j, \forall j \in \mathcal{N} \\
& 0 \le \lambda_{ji} \le a_i, \forall i \ne j \in \mathcal{N} \\
& a_j + b_j \le 1, \forall j \in \mathcal{N} \\
& a_j, b_j \in \{0, 1\}, \forall j \in \mathcal{N}.
\end{aligned}
\tag{6}
$$

Here $\mathbf{f}, \mathbf{g}$ are two parameter vectors with non-negative components. Data point $\mathbf{x}_i$ is selected if $a_i = 1$. Data point $\mathbf{x}_j$ is contained in the convex combination of selected data points if $b_j = 1$. When $\epsilon' = 0$, for any $j \in \mathcal{N}$ with $b_j = 1$, the first two constraints express $\mathbf{x}_j$ as the convex combination of points in $\{\mathbf{x}_i \mid \lambda_{ji} > 0\}$. Here we introduce a small constant $\epsilon'$ to allow some perturbation. The third inequality $0 \le \lambda_{ji} \le a_i$ means we can only use selected data points, which are those with $a_i = 1$, to express other data points. The last constraint $a_j + b_j \le 1$ ensures that any selected data point cannot be expressed as a convex combination of other selected data points. Depending on the choice of $\mathbf{f}, \mathbf{g}$ and $\epsilon'$, we have many different variants of (6). In the next section, we describe one variant of data-selection. We discuss *balanced* data-selection in the supplementary material.

## 3.1 Selecting approximal extreme points

We notice that the original 0-1 LP can be formulated to maximize the number of data points inside the convex hull of selected data points by selecting $\mathbf{f} = \mathbf{0}$ and $\mathbf{g} = \mathbf{1}$. This special case of (6) is used because choosing these values allows us to decompose it into $N$ smaller LPs while maximizing the points inside the convex hull. By projecting out variable $\mathbf{a}$, the resulting 0-1 LP is equivalent to the following LP, as shown by the next result:

$$
\begin{aligned}
\max \quad & \sum_{i \in \mathcal{N}} b_i \\
\text{s.t.} \quad & -\epsilon' \cdot \mathbf{1} \le b_j \mathbf{x}_j - \sum_{i \ne j} \lambda_{ji} \mathbf{x}_i \le \

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

# A Scalable Mixed-Integer Programming Based Framework for Optimal Decision Trees Supplementary Material, NeurIPS 2020

## A Supplementary material for Section 2

**Theorem 1.** *Every integer variable $u_l, l \in \mathcal{L}$ in (1) can be relaxed to be continuous in $[1, Y]$.*

*Proof of Theorem 1.* It suffices to show, for any extreme point $V$ of $\mathrm{conv}(\bar{P})$, where $\bar{P}$ is the feasible region in (1) while all variables $u_l$ are relaxed to $[1, Y]$, the $\mathbf{u}$ components of $V$ are still integer. We denote $V(u_i)$ to be the $u_i$ component in extreme point $V$, analogously we have $V(c_i), V(h_{b,j})$ and so on.

Arbitrarily pick $l \in \mathcal{L}$. If for all $i \in [n], V(e_{il}) = 0$, meaning there is no data point entering leaf node $l$, then we can see that $V(u_l)$ can be any value in $[1, Y]$ and this would maintain the feasibility of point $V$. Since $V$ is an extreme point, $V(u_l)$ has to be either $1$ or $Y$ in this case, which are both integer. Now we assume there exists some $i \in [n]$ such that $V(e_{il}) = 1$, then from (1c)-(1e), we see that $V(u_l) = V(w_{il}) = V(\hat{y}_i)$. From (1b): $(y_i - Y)V(c_i) \leq y_i - V(\hat{y}_i) \leq (y_i - 1)V(c_i)$, here $V(c_i) \in \{0, 1\}$, we get: when $V(c_i) = 0$, then $V(\hat{y}_i) = y_i$, which implies $V(u_l) = y_i$, it is an integer. If $V(c_i) = 1$, then (1b) degenerates to $V(\hat{y}_i) \in [1, Y]$. From the extreme point assumption on $V$, we know in this case $V(\hat{y}_i)$ should be either $1$ or $Y$, then from $V(u_l) = V(\hat{y}_i)$, we know $V(u_l)$ is still integral. $\square$

**Theorem 2.** *Let $(\boldsymbol{h}'_b, g'_b)_{b \in \mathcal{B}}, (u'_l)_{l \in \mathcal{L}}, (\hat{y}'_i, c'_i)_{i \in [n]}, (w'_{il}, e'_{il})_{i \in [n], l \in \mathcal{L}}, (p_{ib}^{+\prime}, p_{ib}^{-\prime}, m'_{ib})_{i \in [n], b \in \mathcal{B}}$ be a feasible solution to (1) with parameters $(\alpha_1, \alpha_2, M, \epsilon)$. Then it is further an optimal solution to (1) with parameters $(\alpha_1, \alpha_2, M, \epsilon)$, if and only if $(\boldsymbol{h}'_b/M, g'_b/M)_{b \in \mathcal{B}}, (u'_l)_{l \in \mathcal{L}}, (\hat{y}'_i, c'_i)_{i \in [n]}, (w'_{il}, e'_{il})_{i \in [n], l \in \mathcal{L}}, (p_{ib}^{+\prime}/M, p_{ib}^{-\prime}/M, m'_{ib}/M)_{i \in [n], b \in \mathcal{B}}$ is an optimal solution to (1) with parameters $(M\alpha_1, M\alpha_2, 1, \epsilon/M)$.*

*Proof of Theorem 2.* Let $(\mathbf{h}'_b, g'_b)_{b \in \mathcal{B}}, (u'_l)_{l \in \mathcal{L}}, (\hat{y}'_i, c'_i)_{i \in [n]}, (w'_{il}, e'_{il})_{i \in [n], l \in \mathcal{L}}, (p_{ib}^{+\prime}, p_{ib}^{-\prime}, m'_{ib})_{i \in [n], b \in \mathcal{B}}$ be an arbitrary feasible solution to (1) with parameters $(\alpha_1, \alpha_2, M, \epsilon)$. So (1b)-(1e), (1k) hold, and

$$g'_b - \sum_{j \in [d]} h'_{bj} \cdot x_{ij} = p_{ib}^{+\prime} - p_{ib}^{-\prime}, \forall i \in [n], b \in \mathcal{B}$$

$$p_{ib}^{+\prime} \leq M(1 - e'_{il}), \forall i \in [n], l \in \mathcal{L}, b \in A_R(l)$$

$$p_{ib}^{-\prime} + m'_{ib} \geq \epsilon e'_{il}, \forall i \in [n], l \in \mathcal{L}, b \in A_R(l)$$

$$p_{ib}^{-\prime} \leq M(1 - e'_{il}), \forall i \in [n], l \in \mathcal{L}, b \in A_L(l)$$

$$p_{ib}^{+\prime} + m'_{ib} \geq \epsilon e'_{il}, \forall i \in [n], l \in \mathcal{L}, b \in A_L(l),$$

which is equivalent to:

$$g'_b/M - \sum_{j \in [d]} h'_{bj}/M \cdot x_{ij} = p_{ib}^{+\prime}/M - p_{ib}^{-\prime}/M, \forall i \in [n], b \in \mathcal{B}$$

$$p_{ib}^{+\prime}/M \leq 1 - e'_{il}, \forall i \in [n], l \in \mathcal{L}, b \in A_R(l)$$

$$p_{ib}^{-\prime}/M + m'_{ib}/M \geq \epsilon/M e'_{il}, \forall i \in [n], l \in \mathcal{L}, b \in A_R(l)$$

$$p_{ib}^{-\prime}/M \leq 1 - e'_{il}, \forall i \in [n], l \in \mathcal{L}, b \in A_L(l)$$

$$p_{ib}^{+\prime}/M + m'_{ib}/M \geq \epsilon/M e'_{il}, \forall i \in [n], l \in \mathcal{L}, b \in A_L(l),$$

Note that constraints (1b)-(1e), (1k) do not involve variables $(h_b, g_b)_{b \in \mathcal{B}}, (p_{ib}^+, p_{ib}^-, m_{ib})_{i \in [n], b \in \mathcal{B}}$. Therefore, we know that solution $(h_b', g_b')_{b \in \mathcal{B}}$, $(u_l')_{l \in \mathcal{L}}$, $(\hat{y}_i', c_i')_{i \in [n]}$, $(w_{il}', e_{il}')_{i \in [n], l \in \mathcal{L}}$, $(p_{ib}^{+'}, p_{ib}^{-'}, m_{ib}')_{i \in [n], b \in \mathcal{B}}$ is feasible to (1) with parameters $(M, \epsilon)$, if and only if, $(h_b'/M, g_b'/M)_{b \in \mathcal{B}}, (u_l')_{l \in \mathcal{L}}, (\hat{y}_i', c_i')_{i \in [n]}, (w_{il}', e_{il}')_{i \in [n], l \in \mathcal{L}}, (p_{ib}^{+'}/M, p_{ib}^{-'}/M, m_{ib}'/M)_{i \in [n], b \in \mathcal{B}}$ is feasible to (1) with parameters $(1, \epsilon/M)$.

Furthermore, if $(h_b', g_b')_{b \in \mathcal{B}}, \dots$ is the optimal solution to (1) with parameters $(\alpha_1, \alpha_2, M, \epsilon)$, then for any feasible solution $(h_b'', g_b'')_{b \in \mathcal{B}}, \dots$ to (1) with parameters $(M, \epsilon)$, there is

$$\sum_i c_i' + \alpha_1 \sum_{i,b} m_{ib}' + \alpha_2 \sum_{b,j} |h_{bj}'| \leq \sum_i c_i'' + \alpha_1 \sum_{i,b} m_{ib}'' + \alpha_2 \sum_{b,j} |h_{bj}''|.$$

Since $(h_b'/M, g_b'/M)_{b \in \mathcal{B}}, (u_l')_{l \in \mathcal{L}}, (\hat{y}_i', c_i')_{i \in [n]}, (w_{il}', e_{il}')_{i \in [n], l \in \mathcal{L}}, (p_{ib}^{+'}/M, p_{ib}^{-'}/M, m_{ib}'/M)_{i \in [n], b \in \mathcal{B}}$ and $(h_b''/M, g_b''/M)_{b \in \mathcal{B}}, (u_l'')_{l \in \mathcal{L}}, (\hat{y}_i'', c_i'')_{i \in [n]}, (w_{il}'', e_{il}'')_{i \in [n], l \in \mathcal{L}}, (p_{ib}^{+''}/M, p_{ib}^{-''}/M, m_{ib}''/M)_{i \in [n], b \in \mathcal{B}}$ are both feasible to (1) with parameters $(1, \epsilon/M)$, and the objective function of (1) with parameters $(M\alpha_1, M\alpha_2)$ is $\sum_i c_i + M\alpha_1 \sum_{i,b} m_{ib} + M\alpha_2 \sum_{b,j} |h_{bj}|$, therefore:

$$\sum_i c_i' + M\alpha_1 \sum_{i,b} m_{ib}'/M + M\alpha_2 \sum_{b,j} |h_{bj}'|/M$$

$$= \sum_i c_i' + \alpha_1 \sum_{i,b} m_{ib}' + \alpha_2 \sum_{b,j} |h_{bj}'|$$

$$\leq \sum_i c_i'' + \alpha_1 \sum_{i,b} m_{ib}'' + \alpha_2 \sum_{b,j} |h_{bj}''|$$

$$= \sum_i c_i'' + M\alpha_1 \sum_{i,b} m_{ib}''/M + M\alpha_2 \sum_{b,j} |h_{bj}''|/M.$$

Hence $(h_b'/M, g_b'/M)_{b \in \mathcal{B}}, (u_l')_{l \in \mathcal{L}}, (\hat{y}_i', c_i')_{i \in [n]}, (w_{il}', e_{il}')_{i \in [n], l \in \mathcal{L}}, (p_{ib}^{+'}/M, p_{ib}^{-'}/M, m_{ib}'/M)_{i \in [n], b \in \mathcal{B}}$ is an optimal solution to (1) with parameters $(M\alpha_1, M\alpha_2, 1, \epsilon/M)$. The other direction is exactly the same. □

## A.1 Cutting-planes for SVM1-ODT

In this subsection, we provide proofs for Theorem 3 and Proposition 1. Note that we denote by $\mathcal{N}_k$ the set of data points with the same dependent value $k$, i.e., $\mathcal{N}_k := \{i \in [n] : y_i = k\}$, for any $k \in [Y]$.

**Theorem 3.** *Given a set $I \subseteq [n]$ with $|I \cap \mathcal{N}_k| \leq 1$ for any $k \in [Y]$. Then for any $L \subseteq \mathcal{L}$, the inequality*

$$\sum_{i \in I} c_i \geq \sum_{i \in I, l \in L} e_{il} - |L|$$

*is a valid cutting-plane to SVM1-ODT* (1).

Here the index set $I$ is composed by arbitrarily picking at most one data point from each class $\mathcal{N}_k$.

*Proof of Theorem 3.* First, we consider the case that $\sum_{l \in L} e_{il} = 1$ for all $i \in I$. This means for any data point $i \in I$, it is classified into leaf node among $L$. Then inequality (5) reduces to: $\sum_{i \in I} c_i \geq |I| - |L|$. It trivially holds when $|L| \geq |I|$. When $|I| > |L|$, according to the property that $|I \cap \mathcal{N}_k| \leq 1$ for any $k \in [Y]$, we know the data points in $I$ have exactly $|I|$ many different classes. Since all of those data points are entering leaf nodes in $L$, according to the Pigeon Hole Principle, we know at least $|I| - |L|$ of them are misclassified.

Now we consider general case. We can divide $I$ into two parts: $I = I_1 \cup I_2$, where $I_1 := \{i \in I : \sum_{l \in L} e_{il} = 1\}, I_2 := \{i \in I : \sum_{l \in L} e_{il} = 0\}$. Then from our above discussion, we have $\sum_{i \in I_1} c_i \geq \sum_{i \in I_1, l \in L} e_{il} - |L|$. From the definition of $I_2$, there is $\sum_{i \in I_2, l \in L} e_{il} = 0$. Hence:

$$\sum_{i \in I} c_i \geq \sum_{i \in I_1} c_i \geq \sum_{i \in I_1, l \in L} e_{il} - |L|$$

$$= \sum_{i \in I_1, l \in L} e_{il} - |L| + \sum_{i \in I_2, l \in L} e_{il} = \sum_{i \in I, l \in L} e_{il} - |L|.$$

We complete the proof. $\qquad\square$

**Proposition 1**. *Let* $\{|\mathcal{N}_k| \mid k \in [Y]\} = \{s_1, \ldots, s_k\}$ *with* $s_1 \leq s_2 \leq \ldots \leq s_Y$. *Then the following inequalities*

1. $\forall l \in \mathcal{L}, \sum_{i \in [n]} c_i \geq \sum_{i \in [n]} e_{il} - s_Y$;

2. $\forall l \in \mathcal{L}, \sum_{i \in [n]} (c_i + e_{il}) \geq s_1 \cdot (Y - 2^D + 1)$;

3. $\sum_{i \in [n]} c_i \geq s_1 + \ldots + s_{Y-2^D}$ *if* $Y > 2^D$.

*are all valid cutting-planes to SVM1-ODT* (1).

*Proof of Proposition 1.* We prove the validity of these inequalities individually.

1. Since $s_1 \leq \ldots \leq s_Y$, we can partition the set $[n]$ into $s_Y$ different disjoint $I$s that all have property $|I \cap \mathcal{N}_k| \leq 1$ for any $k \in [Y]$ as in Theorem 3. Select $\mathcal{L} = \{l\}$ for an arbitrary $l \in \mathcal{L}$. Then, for each of the above set $I$ we obtain a cut from (5). Combining all these $S_Y$ cuts together, we would obtain the desired inequality $\sum_{i \in [n]} c_i \geq \sum_{i \in [n]} e_{il} - s_Y$.

2. For any $l \in \mathcal{L}$, denote $L := \mathcal{L} \setminus \{l\}$. Then for index set $I$ with $|I \cap \mathcal{N}_k| \leq 1$ for any $k \in [Y]$, Theorem 3 gives us the inequality $\sum_{i \in I} c_i \geq \sum_{i \in I, l' \in L} e_{il'} - |L|$. From (1k) of SVM1-ODT, there is $\sum_{l' \in L} e_{il'} = 1 - e_{il}$. Hence we obtain the inequality

$$\sum_{i \in I} (c_i + e_{il}) \geq |I| - 2^D + 1.$$

Then we construct $s_1$ number of disjoint $I$s each with cardinality $|Y|$, by arbitrarily picking one element from each class $\mathcal{N}_k$ without replacement. This process can be done for exactly $s_1$ many times. Therefore, we obtain $s_1$ many inequalities for different index set $I$:

$$\sum_{i \in I} (c_i + e_{il}) \geq |Y| - 2^D + 1.$$

For those $i \in [n]$ which are not contained in any of these $s_1$ many $I$s, we use the trivial lower bound inequality $c_i + e_{il} \geq 0$. Lastly, we combine all those inequalities together, which leads to:

$$\sum_{i \in [n]} (c_i + e_{il}) \geq s_1 \cdot (Y - 2^D + 1).$$

3. When $Y > 2^D$, since there are more classes than the number of leaf nodes, and all data points have to be categorized into exactly one of those leaf nodes, by the Pigeonhole Principle, we know the number of misclassification is at least $s_1 + \ldots + s_{Y-2^D}$.

$\qquad\square$

## A.2 SVM1-ODT for data-set with Categorical Features

Usually in practice, when dealing with data-set with both numerical and categorical features, people will do some feature transformation in pre-processing, e.g., one-hot encoding etc. It is doing fine for most heuristic-based training methods, but since here our training method is MIP-based, the tractability performance can be very dimension-sensitive. Most feature transformation increases the data dimension greatly. In this subsection we are going to talk about the modifications of

the formulation SVM1-ODT, in order to train the initial data-set directly without doing feature transformation. First we add the following new constraints:

$$\sum_{j \in \mathcal{F}_c} h_{bj} \leq 1, \forall b \in \mathcal{B} \tag{8a}$$

$$s_{bjv} \leq h_{bj}, \forall b \in \mathcal{B}, j \in \mathcal{F}_c, v \in \mathcal{V}_j \tag{8b}$$

$$\sum_{j \in \mathcal{F}_c} \sum_{v \in \mathcal{V}_j} s_{bjv} \mathbb{I}(x_{i,j} = v) \geq e_{il} + \sum_{j \in \mathcal{F}_c} h_{bj} - 1, \tag{8c}$$

$$\forall i \in [n], l \in \mathcal{L}, b \in A_L(l)$$

$$\sum_{j \in \mathcal{F}_c} \sum_{v \in \mathcal{V}_j} s_{bjv} \mathbb{I}(x_{i,j} = v) \leq 2 - e_{il} - \sum_{j \in \mathcal{F}_c} h_{bj}, \tag{8d}$$

$$\forall i \in [n], l \in \mathcal{L}, b \in A_R(l)$$

$$\sum_{j \in \mathcal{F}_c} h_{bj} - 1 \leq h_{bj} \leq 1 - \sum_{j \in \mathcal{F}_c} h_{bj}, \tag{8e}$$

$$\forall b \in \mathcal{B}, j \in \mathcal{F}_q$$

$$s_{bjv}, h_{bj} \in \{0, 1\}, \forall b \in \mathcal{B}, j \in \mathcal{F}_c, v \in \mathcal{V}_j \tag{8f}$$

Here $\mathcal{F}_c \subseteq [d]$ denotes the index set of categorical features, and $\mathcal{F}_q = [d] \setminus \mathcal{F}_c$ denotes the index set of numerical features. $\mathbb{I}(x_{i,j} = v)$ is the indicator function about whether $x_{i,j} = v$ or not. For $j \in \mathcal{F}_c$, $\mathcal{V}_j$ denotes the set of all possible values at categorical feature $j$.

For our above formulation, we are assuming: At each branch node of the decision tree, the branching rule cannot be based on both categorical features and numerical features, and if it is based on categorical feature, then the branching rule is only provided by one single categorical feature. These constraints are formulated by (8a) and (8e) individually, where $h_{bj} \in \{0, 1\}$ for $j \in \mathcal{F}_c$ characterizes whether the branching rule at node $b$ is based on categorical feature $j$ or not. Furthermore, we use another binary variable $s_{bjv}$ to denote the conditions that data points must satisfy in order to go left at that node $b$: If the $j$-th component of data point $\mathbf{x}_i$ has value $v$ such that $s_{bjv} = 1$, then this data point goes to the left branch. Here we allow multiple different values $v \in \mathcal{V}_j$ such that $s_{bjv} = 1$. Clearly $s_{bjv}$ can be positive only when categorical feature $j$ is chosen as the branching feature. So we have constraint (8b). Note that this branching rule for categorical features have also been considered in [1], through different formulation. Now we explain how to formulate this categorical branching rule into linear constraints. Still, we start from the leaf node. When $e_{il} = 1$, we know the branching rule at each $b \in A_L(l) \cup A_R(l)$ is applied to data point $\mathbf{x}_i$. To be specific, when $b \in A_L(l)$ and $\sum_{j \in \mathcal{F}_c} h_{bj} = 1$, there must exists some feature $j \in \mathcal{F}_c$ and $v \in \mathcal{V}_j$ such that $\mathbf{x}_{i,j} = v$ and $s_{bjv} = 1$. Analogously, when $b \in A_R(l)$ and $\sum_{j \in \mathcal{F}_c} h_{bj} = 1$, the previous situation cannot happen. Then one can easily see that these two implications are formulated by (8c) and (8d). Cases of $\sum_{j \in \mathcal{F}_c} h_{bj} = 0$ are not discussed here since in that case the branching rule is given by the branching hyperplane on numerical features, which was discusses in Sect. 2.

Furthermore we should mention that, to train data-set with both numerical and categorical features, we will also modify some constraints in the original formulation (1): (1h) and (1j) should be changed to $p_{ib}^- + m_{ib} \geq \epsilon(e_{il} - \sum_{j \in \mathcal{F}_c} h_{bj})$ and $p_{ib}^+ + m_{ib} \geq \epsilon(e_{il} - \sum_{j \in \mathcal{F}_c} h_{bj})$. This is because those two constraints are only considered when the branching rule is not given by categorical features, which means $\sum_{j \in \mathcal{F}_c} h_{bj} = 0$.

## B Supplementary material for Section 3

**Theorem 4.**

1) If $\epsilon' = 0$, then for any optimal solution $(\boldsymbol{b}, \bar{\lambda})$ of (7), there exists $\lambda$ s.t. $(\boldsymbol{b}, \lambda)$ is optimal solution of (6) with $\boldsymbol{f} = \boldsymbol{0}, \boldsymbol{g} = \boldsymbol{1}$, and vice versa;

2) If $\epsilon' > 0$, then for any optimal solution $(\boldsymbol{b}, \bar{\lambda})$ of (7), there exists $\lambda$ s.t. $(\lfloor \boldsymbol{b} \rfloor, \lambda)$ is an optimal solution of (6) with $\boldsymbol{f} = \boldsymbol{0}, \boldsymbol{g} = \boldsymbol{1}$. Here, $\lfloor \boldsymbol{b} \rfloor$ is a vector with every component being $\lfloor b_i \rfloor$.

*Proof of Theorem 4.* We consider two cases for $\epsilon'$.

**Case 1** ($\epsilon' = 0$): First we show: If $(\mathbf{b}, \bar{\lambda})$ is an optimal solution in (7), then there exists $\lambda$ such that $(\mathbf{b}, \lambda)$ is optimal in (6). Notice that when $\epsilon' = 0, \mathbf{b} \in \{0,1\}^{|\mathcal{N}|}$. This is because if $b_j \in (0,1)$, we can multiply $b_j$ and all $\lambda_{ji}$ by $\frac{1}{b_j}$, the new solution is still feasible, with higher objective value, which contradicts to the optimal assumption of $\mathbf{b}$. Assuming there exists some $i \in \mathcal{N}$ such that $b_i = 1$ and exists some $j \neq i \in \mathcal{N}$ such that $\bar{\lambda}_{ji} > 0$. Since $b_j = \sum_i \bar{\lambda}_{ji}$, we know $b_j = 1$. Now we transform the original $j$-th row of $\bar{\lambda}$ in this way: replace $\bar{\lambda}_{ji}$ by 0, replace $\bar{\lambda}_{ji'}$ by $\frac{\bar{\lambda}_{ji}\bar{\lambda}_{ii'}+\bar{\lambda}_{ji'}}{1-\bar{\lambda}_{ji}\bar{\lambda}_{ij}}$. Here $i' \neq i, j$. As long as there exists $i \neq j \in \mathcal{N}$ such that $b_i = 1, \lambda_{j,i} > 0$, we can do the above transformation to get rid of it. Therefore after doing this transformation for finitely many times, we obtain a new $\lambda$ with $0 \leq \lambda_{ji} \leq 1 - b_i, \forall i \neq j \in \mathcal{N}$. It remains to show, after each transformation, the first two equations in (7) still hold. The first equation holds because $\mathbf{x}_j = \bar{\lambda}_{ji}\mathbf{x}_i + \sum_{i'\neq j,i} \bar{\lambda}_{ji'}\mathbf{x}_{i'} = \bar{\lambda}_{ji}(\sum_{i'\neq i} \bar{\lambda}_{ii'}\mathbf{x}_{i'}) + \sum_{i'\neq j,i} \bar{\lambda}_{ji'}\mathbf{x}_{i'} = \bar{\lambda}_{ji}\bar{\lambda}_{ij}\mathbf{x}_j + \sum_{i'\neq j,i}(\bar{\lambda}_{ji}\bar{\lambda}_{ii'} + \bar{\lambda}_{ji'})\mathbf{x}_{i'}$. Hence $\mathbf{x}_j = \sum_{i'\neq j,i} \frac{\bar{\lambda}_{ji}\bar{\lambda}_{ii'}+\bar{\lambda}_{ji'}}{1-\bar{\lambda}_{ji}\bar{\lambda}_{ij}}\mathbf{x}_{i'}$. For the second equation we just need to check $1 = \sum_{i'\neq j,i} \frac{\bar{\lambda}_{ji}\bar{\lambda}_{ii'}+\bar{\lambda}_{ji'}}{1-\bar{\lambda}_{ji}\bar{\lambda}_{ij}}$. This is true because $\sum_{i'\neq j,i} (\bar{\lambda}_{ji}\bar{\lambda}_{ii'} + \bar{\lambda}_{ji'})/(1 - \bar{\lambda}_{ji}\bar{\lambda}_{ij}) = (\sum_{i'\neq j,i}(\bar{\lambda}_{ji}\bar{\lambda}_{ii'} + \bar{\lambda}_{ji'}))(1 - \bar{\lambda}_{ji}\bar{\lambda}_{ij}) = (\bar{\lambda}_{ji}(\sum_{i'\neq j,i} \bar{\lambda}_{ii'}) + \sum_{i'\neq j,i} \bar{\lambda}_{ji'})(1 - \bar{\lambda}_{ji}\bar{\lambda}_{ij}) = (\bar{\lambda}_{ji}(1 - \bar{\lambda}_{ij}) + (1 - \bar{\lambda}_{ji}))(1 - \bar{\lambda}_{ji}\bar{\lambda}_{ij}) = 1$. Therefore, we have shown for any optimal solution $(\mathbf{b}, \bar{\lambda})$ of (7), there exists $\lambda$ s.t. $(\mathbf{b}, \lambda)$ is feasible in (6). The optimality of such $(\mathbf{b}, \lambda)$ in (6) automatically follows from the fact that (7) is a relaxation over (6). Now, we show the other direction. It suffices to show: If $(\mathbf{b}, \lambda)$ is an optimal solution of (6), then $(\mathbf{b}, \lambda)$ is also an optimal solution of (7). Clearly $(\mathbf{b}, \lambda)$ is feasible in (7). If it is not the optimal solution, meaning there exists $(\mathbf{b}', \bar{\lambda})$ in (7), such that $\mathbf{1}^T\mathbf{b}' > \mathbf{1}^T\mathbf{b}$. From what we just showed, there exists $\lambda'$ such that $(\mathbf{b}', \lambda')$ is feasible in (6), with $\mathbf{1}^T\mathbf{b}' > \mathbf{1}^T\mathbf{b}$, which gives the contradiction.

**Case 2** ($\epsilon' > 0$): Since $\lfloor \mathbf{b}^* \rfloor$ is a binary vector, follow the same argument as in above, we can construct $\lambda$ such that $(\lfloor \mathbf{b}^* \rfloor, \lambda)$ is a feasible solution to (6). To show it is also optimal, it suffices to show, any feasible solution in (6), has objective value no larger than $\sum_{i \in \mathcal{N}} \lfloor b_i^* \rfloor$. Assume $(b_i)_{i \in \mathcal{N}}$ is part of one feasible solution in (6). Then apparently it is also part of one feasible solution in (7), since (7) is simply the relaxation of (6). Now we show $b_i \leq b_i^*$ for all $i$: when $b_i = 0$ it is trivially true, when $b_i = 1$ then $b_i^*$ should also be 1, since otherwise we can also increase $b_i^*$ to 1 while remain feasible, and this contradicts to the optimality assumption of $\mathbf{b}^*$. Hence: $\sum_i \lfloor b_i \rfloor \leq \sum_i \lfloor b_i^* \rfloor$. Since $\mathbf{b}$ is binary, then $\sum_i \lfloor b_i \rfloor = \sum_i b_i$, and this concludes the proof. □

## B.1 Balanced Data-selection

In this section we talk about another variant of (7), under the same data-selection framework as we mentioned in Sect. 3, called *balanced data-selection*. Sometimes when doing data-selection, it might not be a good idea to pick all the extreme points or discard as many points inside the convex hull as possible. One example is Figure 3: In this data-set, most of data points lie inside the rectangle, some of them lie on the circle, and a few of them lie inside the circle while outside the rectangle. If we only care about maximizing the number of points inside the convex hull of entire data points, then we would end up picking all points on the circle, however in this case it is obvious that picking those 4 vertices of the rectangle is a better option since they capture the data distribution better, even though these 4 points are not extreme points of the entire data-set.

Motivated by the above example, we realize that in many cases the balance between the number of selected points (those with $a_i = 1$ in (6)) and the number of points inside the convex hull of selected points (points with $b_i = 1$) is important. This can be represented by choosing nonzero objective $\mathbf{f}$ and $\mathbf{g}$. One obvious choice here is $\mathbf{f} = \mathbf{g} = \mathbf{1}$. However, once we pick nonzero objective $\mathbf{f}$, we cannot simply project out $\mathbf{a}$ variables by replacing $a_i = 1 - b_i$ any more, and 0-1 LP (6) might also seem intractable to solve optimally in large data-set case. But since this 0-1 LP was proposed by heuristic, we do not have to solve it to optimality. Here we can approximately solve it by looking at the optimal solution of its corresponding linear

Figure 3: In this case, the balanced data-selection is more preferable.

relaxation:

$$\begin{aligned}
\max \quad & \sum_{i \in \mathcal{N}} (b_i - a_i) \\
s.t. \quad & b_j \mathbf{x}_j = \sum_{i \in \mathcal{N}, i \neq j} \lambda_{ji} \mathbf{x}_i, \forall j \in \mathcal{N} \\
& \sum_{i \in \mathcal{N}, i \neq j} \lambda_{ji} = b_j, \forall j \in \mathcal{N} \\
& 0 \leq \lambda_{ji} \leq a_i, \forall i \neq j \in \mathcal{N} \\
& a_j + b_j \leq 1, \forall j \in \mathcal{N} \\
& a_j, b_j \in [0, 1].
\end{aligned} \tag{9}$$

(9) is the natural linear relaxation for (6) with $\epsilon' = 0$. Then we have the following observations:

**Remark 1.** *Assume $(\boldsymbol{a}', \boldsymbol{b}', \lambda')$ is the optimal solution of* (9). *Then:*

- *If $b_i' = 0$ for some $i \in \mathcal{N}$, then $\boldsymbol{x}_i$ cannot be written as the convex combination of some other points;*

- *If $a_i' = 0$ for some $i \in \mathcal{N}$, then $b_i' \in \{0, 1\}$;*

- *If $a_i' > 0, b_i' > 0$ for some $i \in \mathcal{N}$, then $a_i' + b_i' = 1$.*

Hence we know that, by enforcing $b_i = 0$ in (6) for those $i$ with $b_i' = 0$, it will not change the optimal solution; For those points $\mathbf{x}_i$ with $b_i' = 1$ and $\mathbf{x}_j$ with $b_j' \in (0, 1)$, we expect that it is easier to express $\mathbf{x}_i$ as the convex combination of other points than $\mathbf{x}_j$, so we greedily assign $b_i = 1, a_i = 0$ in (6). The tricky part is about those point with $a_i' > 0, b_i' > 0$, which is used to express other points and can also be expressed by other points in the optimal solution of (9). So we implement the original 0-1 LP (6) to handle those "ambiguous" points. In other words, we will do the following steps:

---

**Algorithm 2** Approximately get a balanced data-selection solution

---

**Solve** the LP relaxation (9), and denote $(\mathbf{a}', \mathbf{b}', \lambda)$ to be the optimal solution, $I_0 := \{i \in \mathcal{N} : b_i' = 0\}$, $I_1 := \{i \in \mathcal{N} : b_i' = 1\}$;
**Assign** $b_i = 0$ for $i \in I_0$, $b_i = 1$ for $i \in I_1$ in (6) with $\mathbf{f} = \mathbf{g} = \mathbf{1}$, and solve it;
**Pick** the index support of the $\mathbf{a}$ components in the optimal solution as the selected data subset.

---

### B.2 Iterative ODT training algorithm

Provided with the framework of using MIP to train ODT and from the nature of our data-selection method, in this subsection we want to propose a generic iterative method to continuously obtain more accurate ODTs from subsets of data-set. We are introducing this iterative training method corresponding to (7), where $\mathbf{f} = \mathbf{0}, \mathbf{g} = \mathbf{1}$. Note that the balanced data-selection can also be applied to this iterative method. We should also mention that this iterative method shares the same spirit as the classic E/M algorithm in statistics.

---

**Algorithm 3** Iterative ODT training method based on data-selection

---

**Initialize** data-selection parameter $\epsilon'$, SVM1-ODT training parameter $\alpha_1, \alpha_2, \epsilon, M$, and initial heuristically generated decision tree $T$;
**for** $t = 0, 1, 2, \dots$ **do**
  Cluster $\mathcal{N}$ is picked as: the data points of $[n]$ assigned into each leaf node of $T$, which are also been **correctly classified**. Denote the collection of **incorrectly classified data** to be $I$;
  **Solve** the data-selection sub-problem (7) for each cluster $\mathcal{N}$. Denote $\bar{I}_{\mathcal{N}} := \{j \in \mathcal{N} : b_j = 1\}$, $J_{\mathcal{N}} := \{i \notin \bar{I}_{\mathcal{N}} : \exists j \in \bar{I}_{\mathcal{N}}, \text{ s.t. } \lambda_{ji} > 0\}$, $K_{\mathcal{N}} := \mathcal{N} \setminus (\bar{I}_{\mathcal{N}} \cup J_{\mathcal{N}})$. Then denote $J$ to be the collection of all $J_{\mathcal{N}}$, and $K$ to be the collection of all $K_{\mathcal{N}}$;
  **Input** the data subset $I \cup J \cup K$ into MIP (1), and replacing the term $\sum_{i \in I \cup J \cup K} c_i$ in objective function by: $\sum_{j \in J} (|I| + 1) c_j + \sum_{i \in I \cup K} c_i$, and solve the SVM1-ODT;
  **Assign** $T$ to be decision tree corresponding to the optimal (best current feasible) solution of the previous SVM1-ODT, and iterate the loop.
**end for**

---

For this iterative method, we have the following statement.

**Proposition 2.** *When $\alpha_1 = \alpha_2 = 0$, and the SVM1-ODT in each iteration of Algorithm 3 is solved to optimality, then the decision tree obtained at each iteration would have higher accuracy over the entire training data than the previous decision trees.*

*Proof.* For decision tree $T_0$, denote $I$ to be the collection of incorrectly classified data in $T_0$. After solving the data-selection sub-problem (7) for each cluster, we denote $\bar{I}$ to be the collection of data point $i$ in each cluster which has $b_i = 1$, and $J, K$ as defined in the Algorithm 3. Clearly $[n] = I \cup \bar{I} \cup J \cup K$, and the accuracy of $T_0$ is $1 - \frac{|I|}{n}$. Now we assume $T_1$ to be the tree obtained at the next iteration, which minimizes the objective $\sum_{j \in J}(|I|+1)c_j + \sum_{i \in I \cup K} c_i$. Since $T_0$ is a feasible solution, with objective value $|I|$, then we must have, for decision tree $T_1$, $c_j = 0$ for all $j \in J$, and $\sum_{i \in I \cup K} c_i \leq |I|$. In other words, tree $T_1$ correctly classifies all data points in $J$, and the misclassification number over $I \cup K$ is at most $|I|$ many. According to the construction of $J$ and $\bar{I}$, we know that once all data points in $J$ are correctly classified, then all data points in $\bar{I}$ are also correctly classified, since data point in $\bar{I}$ is contained in the convex hull of points in $J$. Also because $[n] = I \cup \bar{I} \cup J \cup K$, we know the training accuracy of $T_1$ over $[n]$ is just $1 - \frac{\sum_{i \in I \cup K} c_i}{n}$, which is at least $1 - \frac{|I|}{n}$, the accuracy of $T_0$. $\square$

## C   Additional numerical results and detailed records

Table 3: Accuracy and running time on medium-sized data-sets, for tree depth D=2. The numbers after '/' for CART and OCT-H are the numerical results reported in [4].

| data-set | Banknote-authen | Echocar-diogram | Seeds | Dermat-ology | Indian-liver | Parkinsons |
|---|---|---|---|---|---|---|
| $n$ | 1372 | 61 | 210 | 358 | 579 | 195 |
| $d$ | 4 | 10 | 7 | 34 | 10 | 21 |
| $Y$ | 2 | 2 | 3 | 6 | 2 | 2 |
| testing accuracy (%) | | | | | | |
| S1O | 99.7 | **100** | **98.7** | **80.7** | **75.2** | **91.0** |
| CART | 89.9 /89.0 | 91.1 /74.7 | 88.9 /87.2 | 65.2 /65.4 | 71.6 /71.7 | 79.9 /84.1 |
| OCT-H | 88.9 /91.5 | 91.1 /77.3 | 88.2 /90.6 | 74.6 /74.2 | 72.3 /72.6 | 86.8 /84.9 |
| Fair | **100** | 86.7 | 90.2 | 73.9 | 71.9 | 81.3 |
| training accuracy (%) | | | | | | |
| S1O | **100** | **100** | **100** | **81.1** | 79.6 | **100** |
| CART | 91.7 | **100** | 92.5 | 67.0 | 71.4 | 88.2 |
| OCT-H | 87.2 | **100** | 94.2 | 75.9 | 75.4 | 92.0 |
| Fair | **100** | **100** | **100** | **81.1** | **81.3** | **100** |
| running time ($s$) | | | | | | |
| S1O | 900 | 5.1 | 39.3 | 900 | 900 | 900 |
| OCT-H | 900 | 0.09 | 519 | 900 | 900 | 900 |
| Fair | 94.8 | 0.28 | 55 | 231 | 600 | 19.6 |

Table 4: Accuracy and running time on medium-sized data-sets, for tree depth D=2. The numbers after '/' for CART and OCT-H are the numerical results reported in [4].

| data-set | sonar | survival | Hepatitis | Relax |
|---|---|---|---|---|
| $n$ | 208 | 306 | 80 | 182 |
| $d$ | 60 | 3 | 19 | 12 |
| $Y$ | 2 | 2 | 2 | 2 |
| testing accuracy (%) | | | | |
| S1O | **82.4** | **75.1** | **89.5** | **73.6** |
| CART | 69.3 /70.4 | 73.4 /73.2 | 80.1 /83.0 | 68.2 /71.1 |
| OCT-H | 74.6 /70.0 | 73.3 /73.0 | 84.2 /81.0 | 66.0 /70.7 |
| Fair | 70.6 | 74.1 | 84.2 | 73.3 |
| training accuracy (%) | | | | |
| S1O | **100** | 74.2 | **100** | 74.9 |
| CART | 79.4 | 77.1 | 94.5 | 74.7 |
| OCT-H | 89.0 | 74.7 | 99.1 | 77.5 |
| Fair | **100** | **80.0** | **100** | **90.5** |
| running time ($s$) | | | | |
| S1O | 900 | 900 | 44 | 900 |
| OCT-H | 900 | 900 | 107 | 900 |
| Fair | 3.14 | 900 | 0.3 | 900 |

Table 5: Testing results for multivariate ODT on medium-sized data-sets, D= 2. The numbers after '/' for CART and OCT-H are the numerical results reported in [4].

| data-set | Balance-scale | Ionosphere | Monks-1 | Monks-2 | Monks-3 | Wine |
|---|---|---|---|---|---|---|
| $n$ | 625 | 351 | 124 | 169 | 122 | 178 |
| $d$ | 4 | 34 | 6 | 6 | 6 | 13 |
| $Y$ | 3 | 2 | 2 | 2 | 2 | 3 |
| testing accuracy (%) | | | | | | |
| S1O | 89.1 | **90.3** | **94.7** | 71.4 | **96.7** | **95.3** |
| CART | 67.5 /64.5 | 87.7 /87.8 | 68.4 /57.4 | 54.8 /60.9 | 94.2 /94.2 | 83.7 /81.3 |
| BinOCT | 69.3 | 88.6 | 80.0 | 58.1 | 93.5 | 90.7 |
| OCT-H | 85.3 /87.6 | 86.4 /86.2 | 90.6 /93.5 | **72.2** /75.8 | 91.8 /92.3 | 88.4 /91.1 |
| Fair | **89.2** | 85.4 | 92.3 | 62.3 | 83.5 | 93.2 |
| training accuracy (%) | | | | | | |
| S1O | 86.5 | **100** | 98.1 | 87.7 | 92.4 | **100** |
| CART | 71.7 | 91.0 | 76.8 | 65.2 | 93.5 | 94.1 |
| BinOCT | 73.3 | 91.1 | 83.5 | 69.8 | 93.8 | 97.3 |
| OCT-H | 82.9 | 94.2 | 94.9 | 79.2 | 90.0 | 97.0 |
| Fair | **89.8** | **100** | **100** | **96.2** | **97.4** | **100** |

Table 6: More testing results for medium-sized data-sets, D=2. Here "S1O-DS" refers to the combination of SVM1-ODT and LP-based data-selection method.

| data-set | statlog | spambase | Thyroid-disease-ann-thyroid | Wall-following-robot-2 | seismic |
|---|---|---|---|---|---|
| $n$ | 4435 | 4601 | 3772 | 5456 | 2584 |
| $d$ | 36 | 57 | 21 | 2 | 20 |
| $Y$ | 6 | 2 | 3 | 4 | 2 |
| testing accuracy (%) | | | | | |
| S1O-DS | **74.2** | **89.1** | 96.3 | 93.7 | 93.3 |
| CART | 63.7 /63.2 | 84.3 /84.2 | **97.5** /95.6 | 93.7/94.0 | 93.3 /93.3 |
| OCT-H | 63.7 /63.2 | 87.2 /85.7 | 92.8 /92.5 | 93.7/94.0 | 93.3 /93.3 |
| Fair | 63.7 | 86.1 | 93.0 | 93.7 | 93.3 |
| training accuracy (%) | | | | | |
| S1O-DS | **74.1** | 88.9 | 96.7 | 93.7 | 93.3 |
| CART | 63.5 | 86.7 | **98.4** | 93.7 | 93.3 |
| OCT-H | 63.5 | 90.3 | 94.2 | 93.7 | 93.3 |
| Fair | 63.5 | **91.4** | 95.8 | 93.7 | 93.3 |
| running time ($s$) | | | | | |
| S1O-DS | 900 | 900 | 900 | 2.08 | 0.16 |
| percentage of selected data points (%) | | | | | |
| DS | 7.0 | 10.0 | 12 | 1.0 | 10.0 |
| Parameters set | | | | | |
| $\epsilon$ | 0.1 | 0.02 | 0.01 | 0.0008 | 0.01 |
| $\alpha_1$ | 1000000 | 1000 | 1000 | 30 | 1000 |
| $\alpha_2$ | 0.1 | 0.1 | 0.1 | 0.01 | 0.1 |
| $\epsilon'$ | 0.0 | 0.0 | 0.0 | 0.0 | 0.0 |
| $\beta_1$ | 0.3 | 0.3 | 0.1 | 0.4 | 0.4 |
| $\beta_2$ | 0.07 | 0.1 | 0.12 | 1.0 | 0.1 |

Table 7: Testing results for large-scale data-sets, with tree depth $D = 2$, time limit is set to be $4h$.

| data-set | Avila | EEG | HTRU | pendigits | skin-segmentation | shuttle |
|---|---|---|---|---|---|---|
| $n$ | 10430 | 14980 | 17898 | 7494 | 245057 | 43500 |
| $d$ | 10 | 14 | 8 | 16 | 3 | 9 |
| $Y$ | 12 | 2 | 2 | 10 | 2 | 7 |
| testing accuracy (%) | | | | | | |
| S1O-DS | **52.6** | **66.5** | **97.8** | **38.9** | **86.3** | **94.0** |
| CART | 50.3 | 58.6 | 97.3 | 36.2 | 80.6 | 93.8 |
| OCT-H | N/A | 58.6 | 97.3 | 36.2 | N/A | N/A |
| training accuracy (%) | | | | | | |
| S1O-DS | **55.0** | **67.1** | 97.5 | **38.9** | **87.1** | **94.1** |
| CART | 50.7 | 60.3 | **97.8** | 36.5 | 81.0 | 94.0 |
| OCT-H | N/A | 60.3 | **97.8** | 36.5 | N/A | N/A |
| percentage of selected data points (%) | | | | | | |
| DS | 4.5 | 2.0 | 3 | 7.1 | 0.42 | 2.0 |
| Parameters set | | | | | | |
| $\epsilon$ | 0.01 | 0.04 | 0.5 | 0.01 | 0.03 | 0.007 |
| $\alpha_1$ | 1000 | 1000 | 1000 | 1000 | 1000 | 1000 |
| $\alpha_2$ | 0.1 | 0.1 | 0.1 | 0.1 | 0.1 | 0.1 |
| $\epsilon'$ | 0.1 | 0.0 | 0.0 | 0.3 | 0.0 | 0.0 |
| $\beta_1$ | 0.2 | 0.1 | 0.1 | 0.1 | 0.2 | 0.1 |
| $\beta_2$ | 0.05 | 0.02 | 0.03 | 0.1 | 0.05 | 0.02 |

Table 8: Testing results for large-scale data-sets, with tree depth $D = 3$, time limit is set to be $4h$.

| data-set | Avila | EEG | HTRU | pendigits | skin-segmentation | shuttle |
|---|---|---|---|---|---|---|
| $n$ | 10430 | 14980 | 17898 | 7494 | 245057 | 43500 |
| $d$ | 10 | 14 | 8 | 16 | 3 | 9 |
| $Y$ | 12 | 2 | 2 | 10 | 2 | 7 |
| testing accuracy (%) | | | | | | |
| S1O-DS | **55.8** | **66.5** | 97.9 | **62.5** | **94.9** | 99.5 |
| CART | 53.5 | 64.2 | **98.1** | 57.9 | 87.1 | **99.7** |
| OCT-H | N/A | N/A | N/A | 57.9 | N/A | N/A |
| training accuracy (%) | | | | | | |
| S1O-DS | **58.1** | **70.6** | 97.4 | **62.2** | **94.7** | 99.5 |
| CART | 56.2 | 65.3 | **97.8** | 58.4 | 87.6 | **99.7** |
| OCT-H | N/A | N/A | N/A | 58.4 | N/A | N/A |
| percentage of selected data points (%) | | | | | | |
| DS | 2.0 | 2.0 | 1.8 | 2.8 | 0.39 | 1.5 |
| Parameters set | | | | | | |
| $\epsilon$ | 0.02 | 0.02 | 0.02 | 0.01 | 0.03 | 0.008 |
| $\alpha_1$ | 1000 | 1000 | 100000 | 1000 | 1000 | 1000 |
| $\alpha_2$ | 0.1 | 0.1 | 0.01 | 0.1 | 0.1 | 0.1 |
| $\epsilon'$ | 0.0 | 0.0 | 0.1 | 0.3 | 0.0 | 0.0 |
| $\beta_1$ | 0.2 | 0.1 | 0.2 | 0.1 | 0.2 | 0.1 |
| $\beta_2$ | 0.02 | 0.02 | 0.04 | 0.08 | 0.03 | 0.015 |