[Reviews · NeurIPS 2020]

Review 1

Summary and Contributions: The paper proposes a novel MIP formulation to train (shallow) decision trees along with an original way to select representatives from large data sets in order to reduce the training set size a priori without sacrificing much accuracy. The MIP formulation is strengthened by a few valid inequalities. Numerical experiments show advantages w.r.t. heuristic (CART) and other MIP approaches. While the proposed methods are very interesting and the results look promising, the paper unfortunately fails to answer some significant questions in the experimental part. Moreover, the Appendix is, in fact, longer than the main paper and contains most of the interesting technical contributions (proofs and methodology) as well as potentially relevant details that are completely omitted from the main paper. Therefore, despite the contributions being interesting, I only score this submission as marginally below the acceptance threshold -- I think it does require a thorough revision to provide a more complete picture and fix some issues, but even more so, this should be submitted as a full-length (journal) paper somewhere instead of trying to squeeze it into NeurIPS's 8 page form. *** update after authors' response *** I thank the authors for their rebuttal. While I don't agree on everything, and the authors do not seem willing to consider/include some aspects (like more comprehensive MIP model comparisons in the experiments, esp. regarding the use of the data selection procedure beforehand) raised by several reviewers, I trust the paper will be improved sufficiently during a revision to slightly raise its quality. Therefore I upgrade my review score from marginally below to marginally above the acceptance threshold. The authors should elaborate more on some related work for their data selection scheme; I forgot this in my initial review, but the field of "exemplar selection" is related and context to methods from that field is needed to clarify novelty of the proposed method.

Strengths: The novel MIP formulation with the proposed enhancements (choice of Big-M constant (possibly) and cutting planes) does itself seem to improve over previous state-of-the-art MIP models to learn optimal decision trees (although such formulations are not directly comparable as they technically solve different problems). The main strength of the paper might, however, be the method to preselect a subset of the data to use for training, in order to facilitate training for very large scale input sets; such preprocessing techniques could have a broader impact beyond the scope of the present work. The theoretical contributions are valuable to the context; I only skimmed over the proofs in the (very long) Appendix but did not see obvious flaws.

Weaknesses: The derivation/explanation of the main model could be clearer at some points. All the import technical content has been delegated to the Appendix, which moreover contains crucial details on several aspects (e.g., the actual data selection algorithm and a heuristic that is apparently being used in place of the proposed 0/1-LP-based method, but was not mentioned once in the "main paper"). The Appendix is longer than the paper itself, which indicates that the paper, while interesting, is not suitable for the page-limited format of NeurIPS and should rather be a full-length journal paper. (This would also give the authors more space for clarifications.) The numerical experiments are promising, but leave open several questions that I consider crucial for a fair and complete discussion of the computational results: How do the other methods (say, CART and OCT-H) perform when trained on the reduced dataset obtained from the data-selection 0/1-LP? Since the optimal ("MIPped") decision trees are very shallow (just 2 or 3 levels), can they still outperform heuristic (e.g., CART) trees if the heuristics are used to create deeper decision trees? When terminating at a time limit, what is the optimality gap of the MIP solver, i.e., how far away from being optimal are the solutions obtained be prematurely terminated MIP models? Are the cutting planes added once at the beginning or are they separated dynamically within the branch-and-cut MIP solving process? Can the cuts also be used to enhance the other MIP models (like OCT-H or "Fair")? Also, the model parameter choices are not discussed -- what are the actual values used for epsilon and the alphas in the experiments? What happens if the actual 0/1-LP method is used rather than the heuristic that the Appendix says was used instead? What different results do the selection scheme outlined in the main paper and the balanced version from the Appendix produce?

Correctness: The supplementary material is longer than the paper itself but contains all proofs and more. I skimmed through the proofs and did not find obvious errors (however, the whole Appendix seems to be fraught with typos). The empirical setup should be extended by experiments/results addressing the points I raised in the previous comment ("Weaknesses") but otherwise seem sound. The main paper should at least mention that all proofs are delegated to the Appendix, and definitely should not lead the reader to believe that the 0/1-LP method is used for the experiments if, as stated in the Appendix, this whole idea is apparently circumvented by simple doing random subsampling!

Clarity: The main paper is mostly written fairly well, the Appendix less so (lots of typos at least). The work nevertheless lacks clarity because several relevant details are moved to the Supplementary part, and some aspects are not mentioned at all (at least in the main paper). The Appendix even contains a section regarding categorical features that is not even hinted at in the main paper. Clarification is needed, e.g., at the following points: - p.2, l.70-73 is too vague, the meaning is unclear -> pls. clarify - p.2, l. 85f: clarify what "[...] $i$ enters leaf node $l$" means (i.e., that data pt. $i$ is "routed" to leaf $l$ through the dec. tree). If $\hat{y}_i$ denotes a predicted label, then why is it real-valued and not in [Y]? (Also regarding the description on p.3, l.96f: why should y_i - \hat{y}_i \geq 1 here -- \hat{y}_i is in R, so couldn't it be, say, y_i - delta for some small delta?) - p.3, l.92: perhaps clarify "tree sparsity" -- actually here this means sparsity of the decision hyperplanes, no the tree itself - The 1-norm is used in the MIP (1) and several times in the text later called "linear" (e.g., p.4, l.136), but this is technically incorrect. As is mentioned only later, it can be *linearized* but is as of itself a nonlinear, nonsmooth function! Also, the linearization described in the text on p.4 (min |z| <=> min t s.t. -t<=z<=t) may not be the best choice -- my experience with a variety of different problems involving the 1-norm is that solvers perform better/faster if the alternative variable-split formulation is used (i.e., replacing z=z^+ - z^- with z^+,z^->=0, so that |z|=z^+ + z^-)! Have you tried this? If not, you should. - p.3, l.108-111: this part is very unclear and needs to be elaborated. In fact, it may be a good idea to move up Section 2.1 entirely, as it is much harder to parse the relatively complicated MIP model without getting the explanations that come there first. - p.4,l.127: you do NOT have a "sparsity condition" here -- instead, sparsity is targeted heuristically by including the 1-norm term in the objective. Similarly, w.r.t. the discussion later (l.144), the 1-norm does NOT "guarantee" model sparsity -- it empirically works well, but actual guarantees are not trivial and often not known. - regarding Thm. 1: why is this not directly incorporated into the MIP (1)? Other aspects like the McCormick envelope are... Also, for Thm. 1 and all other results, there should be at least one mention that proofs are deferred to the Appendix/Supplement! . regarding Thm. 2: It should be clarified that this does not technically eliminate the Big-M problem, it only defers it to the choice of epsilon (and the alphas). So if for actually accurate modeling, one would need a large M and a small epsilon, fixing M=1 then requires epsilon to be chosen even smaller, which itself can introduce numerical issues just as well as a big M. - p.5, l.177f: why would one add cuts randomly? That is almost never a good idea. "Choosing important ones" could (should?) also typically be done in a dynamic fashion, generating and adding violated cuts on the fly during the branch-and-cut solution process. At the very least, here, you should clarify if you add cuts a priori or invoke a separation routine in the MIP solver to find violated cuts to add dynamically! Also, can the cuts be used for the other MIP models as well? If so, this should be done/tested in order to get a fully fair comparison (just like the data preselection routine should not only be combined with the new MIP, but other methods, too!). Moreover, as the cuts only pertain to one part of the objective, an explanation should be added as to why it is guaranteed that if \sum c_I is at its lower bound, the rest (of the variables) is at optimal values, too; I think this is correct, but it may not be clear to everyone why.) - Section 3.1 gives a different scheme (w.r.t. f and g) than I would have expected, given the previous discussion that one wants to max. points in the conv. hull and min. the number of selected data points. This would suggest f=1, g=1 -- which apparently corresponds to the "balanced" scheme that is discussed in the supplement, so that detail should be mentioned (l.216) here to avoid confusion. - Section 3.2: The description here needs to be clarified. First, I suggest swapping I and J notation, as elements of I_N (now) are always referred to as j and those of J_n as i. Then, it seems something went wrong with the sets in the following -- shouldn't J_N contain extreme points to be used for training? Then why choose from K_N? Moreover, what are "boundary hyperplanes $h\in H$" here? This is undefined, and so is the actual meaning of "dist(x_i,h). Finally, the "heuristic" mentioned here does not appear to be the one you mention in the Appendix; in fact, that latter one appears to replace the whole LP machinery you just introduced, so please clarify what is going on here! - p.7, bottom two lines: looking at the numbers, here you probably mean improvement in percentage *points*, not percent? - Table 2 (and others): It would help to give the MIP optimality gap the solvers reached on the different models when they ran into the time limit. - Section 4 could (besides the various different other points mentioned earlier) also include a (brief) discussion of the hyperplane sparsities. You claim your 1-norm term does lead to sparse decision-hyperplane coefficient vectors, as does OCT-H via actual 0-norm, but no evidence for this is provided.

Relation to Prior Work: I found the discussion of prior work adequate. In a full-length paper version (which I think would be the much better format for this work), I would suggest adding a discussion of the differences between the different MIP formulations in some detail, as this is not really done here.

Reproducibility: No

Additional Feedback: Finally, some typos (only in the main paper, there are many more in the Appendix) and other minor points: - Abstract: l.6 "tighten" & l.15 (also p.2, l.52) "heuristic" (remove plural s); - Abstract: the boldface 10% looks odd; same on p.2 -- generally, decide on one way to emphasize things (italics preferably, or bold), but don't mix - p.1, l.25 & p.2, l.49, possibly elsewhere, too: "-" should be a longer dash, "--" in LaTeX - p.1, l.29: "[...] in *the* literature." (missing "the") - p.2, l.48: clarify the type of dec. trees SVM1-ODT is used for (multivar.) - p.2, around l.55: perhaps it should be mentioned that being able to handle only very shallow depths (2 or 3) is arguably the main drawback of MIP-based approaches - p.2, l.63 "comprises" should be "is comprised of" (appears a few more times, I think; e.g., similarly, p.2, l.76) - p.2, l.71: remove semicolon after "selection". - p.2, l.85: remove "is" - p.3, l.101f: one also needs that $u_l$ is bounded (for the McCormick trick to be exact) - could you not simply use $ p^+_{ib}+p^-_{ib}+m_{ib} \geq \epsilon e_{il} \forall l\in\mathcal{L}, b\in\mathcal{B}$ instead of (1h) & (1j)? After all, p^+,p^- perform a classic variable split for the hyperplane deviation via (1f). - The factor 1/2 in front of the 1-norm in (2)-(4) could be removed - p.4, l.125: "branching" - p.4, l.134: "nodes" - p.5, l.163: well, there are cuts to eliminate feasible but suboptimal points, too... - p.5, l.169: "are", not "is" - p.5, l.172: everywhere else, | is used in set definition, here it is : (which I personally prefer anyway) -- stick to one notation throughout - Thm. 3 and Prop. 1: "[...] cutting-planes *for* SVM1-[...]". - p.5, l.194: space missing between period and "For" - p.5, l.200: use a colon instead of a dash - p.6, l.240f: The sentence "The remaining [...] machines." appears left over from some earlier version and must be removed. - p.7,l.267: "intractability" (not "tractability") - p.7,l.268: "the the" <-> "the" - p.8, l.309: "[...] *an* LP-based [...]" - The reference list is inconsistent w.r.t. using full names vs. initials, and it contains multiple typos, formatting errors and other issues. E.g., "O?Sullivan" in [5] or "User?s" in [14], all-lower-cased abbreviations like "Ibm ilog cplex" in [14] or missing and inconsistent capitalization generally (e.g., journal in [7] vs. [15]), odd version/number/"journal" usage as in [14] ("Version" is not a journal name, and the CPLEX 12 is not from 1987) or [26] (the book is "volume 52"...of what?)...


Review 2

Summary and Contributions: The paper develops a model-based approach to train decision trees of limited depth. Specifically, the approach proposes a novel MIP formulation with a l-1 norm minimization criteria that reveals exploitable structure, in that (a) the logical OR constraints deciding tree node branching decisions can be formulated using tighter big-M inequalities, and (b) valid inequalities based on the data structure can be applied to strengthen the LP relaxation. Furthermore, authors propose a separate LP approach to select a subset of the data to use for training. Finally, the paper compares their MIP approach with existing CAT and MIP models from the literature.

Strengths: The paper significant improves upon the state-of-the-art MIP models for designing "optimal" decision trees in terms of scalability with a novel and clever formulation. In particular, the authors provide a "polyhedral perspective" on the structure of the sample that is exploited both in the MIP model (for valid inequalities) as well as in the LP formulation for data selection, which in my view are nice contributions to the field and may be investigated further.

Weaknesses: The scalable MIP formulation and the data selection approach are two distinct stories in the same paper. While the MIP model is well presented and explored, the data selection strategy lacks a proper investigation and seems to be introduced only to artificially reduce the size of the final MIP so that is amenable to larger data samples. Nonetheless, to the best of my understanding, such a method is independent of how the decision tree (or any other ML method) is trained. I was hence left with several open questions concerning its impact (i.e., can it introduce bias? Does it also impact the accuracy of the trained model? etc.), since it is by itself an interest methodology that could lead to a stream of research by itself (more in line with unsupervised learning techniques). Furthermore, model-based approaches for decision trees are still rather limited in their applicability, only feasible to depths of 2 or 3. There is indeed the notion of interpretability and generality of small trees, but results seem to be still far from any practical application, since even for such depths the solution times are rather large.

Correctness: The methodology is correct to the best of my knowledge.

Clarity: The paper is very well written. Perhaps only Section 3 deserves more attention , since it is somewhat dense and the writing itself seems rushed and more difficult to parse, in particular Section 3.2.

Relation to Prior Work: There are two important references missing, which are two of the state-of-the-art model-based techniques for constructing decision trees: Aglin, Gaël, Siegfried Nijssen, and Pierre Schaus. "Learning Optimal Decision Trees Using Caching Branch-and-Bound Search." AAAI. 2020. Verhaeghe, Hélene, et al. "Learning Optimal Decision Trees Using Constraint Programming." BNAIC/BENELEARN. 2019. In particular, I kindly ask the authors to discuss how their methodologies and results compare with the two works above.

Reproducibility: Yes

Additional Feedback: ---------------------------------------------------------- * Response: Thank you for the careful response, they helped me clarify some of my doubts. I believe the paper still needs to better highlight the role of the data selection process, specifically its connection to previous work. ---------------------------------------------------------- - How does CAT perform (in terms of out-of-sample testing) over the same data subset used for the large scale S1O? - Commercial solvers, particularly Gurobi, now implement sophisticated linearization and branching techniques for logical constraints, such as "OR." They do require, however, the use of special constructs to signal the solver that the structure is present. Have the authors attempted those more advanced models? - There are cases where solving the model as a quadratic problem in CPLEX is more effective than using McCormick linearizations. Have the authors attempted that?


Review 3

Summary and Contributions: The authors propose a new mixed integer programming model for obtaining "optimal decision trees" for classification. To improve the convergence behavior, they introduce several valid inequalities to tighten the relaxation. They also propose a sampling scheme for selecting a subset of the dataset. This scheme helps them to alleviate the lack of scalability with solving an integer programming model. To the best of my knowledge, the model and the valid inequalities are new. Though simple, the sampling scheme is promising. ** Update ** I thank the authors for their response. The authors have responded to my comments about lack of comparison against other integer programming based approaches. I agree that their approach is more general but this does not mean that they would not be able to do such a comparison. I believe the data selection part of the paper is more interesting than the MIP model. One of the referees has pointed out the literature on "exemplar selection." I skimmed through some papers on the subject. The topic indeed seems related. This casts a shadow on the novelty of the data selection part. In the light of these two points, I have decided to keep my score.

Strengths: The authors follow the recent work on obtaining optimal decision trees with integer programming. The model and the cuts are interesting. The sampling procedure seems like to have a potential beyond the scope of this paper.

Weaknesses: I have had difficulty to assess the main differences between the proposed mathematical programming model and the models given in [2], [4] and in particular [11] (also the cutting planes). Likewise, there are no numerical experiments comparing their approach against those methods proposed in [2] and [4].

Correctness: The claims are correct. Empirical methodology also sounds correct but the codes are not provided so I cannot reproduce the results.

Clarity: In general the paper is well-written. However there are a few points that I like to mention: 1) How do you define the value of \epsilon in (1h) and (1j)? 2) Line124: "instead of 1 to prevent variables from getting too big." What does 1 stand for here? 3) Line 125 \overline{y}_i : could it be \overhead{y}_i? Otherwise define \overline{y}_i. 4) Line 169: "which are added externally"... 5) Line 194: space is missing "sets.For". 6) Line198: "we merge all the data subsets selected from each leaf node to be the new training data" => ``we select the data points covered by the leaf nodes as the new training data''. 7) Line 348: "User?s" vs User's. 8) Line 387: "All" should be "Every" for consistency... 9) Line 413: Is something missing in parameters here "with parameters (1, \epsilon / M)." here. There are only two parameters here, supposed to be four... 10) Line 497: Should be "multiply". 11) Supplementary Document B.2 should be explained more clearly. Starts good but it is cut very short. 12) Lines 468-482: The use of (9e) and (9a) is not correct. I guess (8a-e) is implied.

Relation to Prior Work: Please see my comments above. To my understanding, the sampling scheme can also be used with other optimal decision tree approaches. If so, the authors could elaborate on this point further.

Reproducibility: No

Additional Feedback:


Review 4

Summary and Contributions: The paper presents a method for learning optimal decision trees from datasets. A mixed integer programming formulation of the problem is developed based on SVMs. It is shown how to efficiently solve this formulation using cutting planes. Finally, a reasonable experimental evaluation of the technique is provided that shows the superiority of the technique.

Strengths: The experimental results reveal that the technique markedly increases the accuracy of predictions made with the decision tree. Compared to previous optimal approaches, the proposed approach significantly increases the size of dataset that can be considered. The data-selection procedure appears to be novel and may be useful to others working on a broader range of problems than the one in the current paper.

Weaknesses: The decision trees studied are rather limited in size. In the largest example, trees of depth 3 are studied. It might be mathematically interesting to look at trees of this size, but in practice much bigger trees are commonly learned (heuristically). While finding single optimal decision trees might be mathematically interesting, much of the machine learning research in the area has moved into working with ensembles of decision trees, e.g. random forests.

Correctness: The MIP formulation appears to be correct. The experimental procedure evaluates against sensible alternatives on commonly used datasets, using sensible metrics. Nothing about the setting appears to offer an unfair advantage or includes surprising choices.

Clarity: The front matter nicely explains the problem to be tackled, the need for work in this area and the overall contributions of the paper. The paper is necessarily very mathematical, but there is some good textual explanation alongside this, such as the description of the MIP model starting at line 94. The experiments are clearly described and the conclusions drawn are valid.

Relation to Prior Work: The paper clearly explains the difference between optimal decision tree learning methods and the more common heursitci methods. There is a reasonable survey of recent MIP approaches to optimal decision tree learning, though this is not in much depth. The novel element of the proposed MIP formulation is clearly explained.

Reproducibility: Yes

Additional Feedback: I didn't follow why the cutting planes listed in proposition 1 are the ones chosen to be added to the problem. Reading between the lines, I'm guessing it is based on preliminary experiments. =================================== UPDATED - I read the other reviews and the authors' response. This has not substantially changed my opinion of the paper.

[Author Response · NeurIPS 2020]

**To Reviewer #1:** *LP-based data selection*: For Fig. 2, we indeed used random sampling to select a subset prior to applying the LP-based data selection on the reduced dataset. This was done because the number of constraints in (7) grows quadratically with the number of samples, $N$. However, upon further analysis we noticed that the random sampling is unnecessary because the LP in (7) with $f = 0$ and $g = 1$ can in fact be decomposed into $N$ smaller LPs, each with $|\mathcal{N}|$ variables and $d + 2$ constraints. These can be solved in parallel using a LP solver. We implemented this for datasets in Fig. 2 and noticed that the performance of our method is no worse than currently reported, and even better in some cases. We thank the reviewer for pointing out this discrepancy, and will update with new experiments for S1O-DS in Fig. 2 with the LP decomposition-based approach (better results without random subsampling preprocessing).

*Cutting planes*: Our cutting planes are added once at the beginning before invoking a MILP solver. The cuts can be applied to the other MIP models provided that they use the variables $c_i$ and $e_{il}$ with the same modelling meaning. For example, Fair in [1] can use these cuts, and OCT-H in [4] needs to be modified slightly before they can be applied.

*Using data selection for CART, OCT-H (also Reviewer #2)*: Thank you for your suggestion. Fig. 1 benchmarks the performance of SVM1-ODT on medium-sized datasets and shows that our ODT formulation outperforms CART, OCT-H, Fair and BinOCT in terms of test accuracy *without using data selection*. Fig. 2 addresses scalability by benchmarking our combined approach (SVM1-ODT and data selection) with CART and other methods such as OCT-H that address scalability. While it is certainly possible to apply our data selection procedure prior to using either CART or OCT-H, our intent in Fig. 2 is largely to benchmark the *scalability* of the combined approach.

*Clarification on Eq. (6)*: In Sec. 3, we first present the generalized data selection formulation that maximizes the points in the convex hull ($g^T b$) and minimizes the number of selected data points ($f^T a$). In Sec. 3.1, we present a special case of (6) with $f = 0$ and $g = 1$ because choosing these values allows us to decompose it into $N$ smaller LPs while maximizing the points inside the convex hull. The balanced schema is presented in the Supp. without numerical results.

*Optimality gap*: For datasets that cannot be solved to optimality within 15 minutes, we observed that the optimality gap could still be large due to the lower bound not improving significantly before the run is terminated. The gap does not indicate time to reach optimality. Thus, we instead use training accuracy to benchmark the performance of our model.

*Notation in Section 3.2*: $J_N$ is a subset of extreme points that satisfy a specific condition. $K_{\mathcal{N}}$ contains extreme points, except they are never used with convex combination coefficient larger than $\frac{1}{d+1}$ to express some other points. Since points in $K_{\mathcal{N}}$ carry some information about the distribution of the dataset, we select a subset of $K_{\mathcal{N}}$, using Alg. 1, for training. "Boundary hyperplanes" refers to the hyperplanes for each leaf node. dist($a$,$h$) refers to the distance between $a$ to the hyperplane $h^T x = 0$. The term "heuristic" refers to Alg. 1. The LP in (7) is solved in the second step of Alg. 1.

*Theorem 2*: Our idea is to disperse the numerical issues between parameters by re-scaling. The numerical instability for a very small $\epsilon$ should be easier to handle by an MILP solver than a very big $M$. *Page 2,line 85*: From (1d,1e), it follows that $w_{il}$ equals to some $u_l$ which is integral. Then (1c) implies that at the optimal solution $\hat{y}_i$ is also integral. So we do not explicitly enforce $\hat{y}_i$ to be integral. *CART with deeper trees*: We follow the standard benchmarking used in [1,4] wherein the tree depth for CART and MIPs are the same. Note that a deeper tree for CART is not necessary to outperform the shallow ODT; e.g., for `Dermatology` the ODT with $D = 2$ has a higher test accuracy (80.7%) compared to the CART tree with $D = 3$ (76.1%) As also observed in [11], for many datasets, even we allow CART to choose the tree depth using its default setting (allowing deep trees), a shallow ODT still outperforms CART (see Tab.13 in [11]). We have incorporated all results in Thm. 1 and 2 into the final model, the final SVM1-ODT model imposes $u_l \in [1, Y]$ and $M = 1$. Model parameters $\epsilon$ and $\alpha_i$ are tuned via cross-validation for each dataset. We will fix the terminology issues and revise discussions related to "sparsity" for hyperplane, "linear" for the 1-norm, and other suggestions.

**To Reviewer #2:** If McCormick linearizations are not used, then there are nonconvex quadratic terms in the constraints. The tractability of the resulting MIQCP is very challenging. CPLEX only supports convex quadratic terms that can be represented as second order cone programs (see shorturl.at/cnDY6). Note that handling logical constraints is not as scalable as the big-M method if we can choose a small value for $M$. As shown in Theorem 2, we can use $M = 1$.

**To Reviewer #3:** The requirements for input features and branching rules from [2] and [11] are different from ours. In particular, [2] and [11] only consider binary features $\mathbf{x}_i \in \{0, 1\}^d$, while our formulation takes numerical features $\mathbf{x}_i \in \mathbb{R}^d$ (and it can be extended to the case with mixed numerical and categorical features, see A.2). Branching rules in these references are binary tests (i.e., univariate splits), while we use a multivariate hyperplane. Hence the authors employ special tools for the specific decision tree, they cannot be generalized to our general tree; for example, [2] proposes a max flow-based MIP formulation. Comparing to [4], we impose additional conditions on hyperplanes so that training samples should be far from the boundary of the cluster at the leaf node by using the multi-hyperplane SVM. Moreover, we use lesser number of binary variables to encode the decision tree ($e_{il}, c_i \in \{0, 1\}$ in our model v.s. $z_{it}, s_{jt}, l_t, d_t \in \{0, 1\}$ in [4]), and we can use a small value for the big-M parameter (i.e., $M = 1$).
We did compare with OCT-H from [4] in Figs 1, 2. The model in [2] handles binary input, and 8 data sets were tested. They share 4 datasets with us (Tab. 2 in [2] and Tab. 5 in our Supp.). For same tree depth, we always outperform [2].
We have defined $\overline{y}_i$ in lines 118-119. For L124, we will replace it by "... a small constant $\epsilon e_{il}$ in (2) instead of $e_{il}$ to ..."

**To Reviewer #4:** We will give more detailed explanation for cutting planes in Prop. 1 in the revised paper.

[Meta-Review · NeurIPS 2020]

This paper is about employing advances in computational efficiency of mixed integer programming methods towards decision tree construction problems. While locally optimal methods can achieve an upper bound on the minimization problem efficiently, closing the optimality gap requires tight lower bounds. The authors use an interval relaxation and a support-vector machine procedure to tighten the lower bound. To scale the algorithm, the authors use a LP-based data selection procedure, and perform all experiments using this procedure. It is not clear whether the global optimality properties of the MIP formulation carry through with the data-selection procedure. Overall, the reviewers viewed the paper as a contribution to the field and the authors responded sufficiently to many of the comments. There is a great deal of interest in developing strong integer programming formulations to problems that were previously thought to be out of reach of MIP solvers and this paper advances the field on its own and suggests further steps for progress.